- Insight into the size-resolved markers and eco-health
- significance of microplastics from typical sources in
- northwest China

- Liyan Liu<sup>1</sup>, Hongmei Xu<sup>1\*,2</sup>, Mengyun Yang<sup>1</sup>, Tafeng Hu<sup>2</sup>, Abdullah Akhtar<sup>1</sup>, Jian Sun<sup>1</sup>, Zhenxing
- Shen<sup>1,2</sup>

- <sup>1</sup>Department of Environmental Science and Engineering, Xi'an Jiaotong University, Xi'an, 710049,
- China
- <sup>2</sup>SKLLQG, Key Lab of Aerosol Chemistry & Physics, Institute of Earth Environment, Chinese
- Academy of Sciences, Xi'an, 710061, China

- \*Corresponding author:
- Hongmei Xu, Department of Environmental Science and Engineering, Xi'an Jiaotong University,
- Xi'an, 710049, China. E-mail: <u>xuhongmei@xjtu.edu.cn</u>.

#### Abstract

Research on atmospheric microplastics (MPs) from typical sources is limited, constraining the targeted management of pollution. Here, the source profiles of eight types of common MPs and three classes of plasticizers (i.e., phthalates, benzothiazole and its derivatives, and bisphenol A) emitted from five living sources: Plastic Burning, Fruit-bag Burning, Road Traffic, Agricultural Film, and Livestock Breeding were determined in PM<sub>2.5</sub> (particulate matter with aerodynamic diameters  $\leq 2.5 \,\mu\text{m}$ ) and PM<sub>10</sub> ( $\leq 10 \,\mu\text{m}$ ) in the Guanzhong Plain, northern China. PB exhibits high proportions of poly(methyl methacrylate) (PMMA) and 2-hydroxy benzothiazole (HOBT), with PMMA being more abundant in PM<sub>2.5-10</sub> (aerodynamic diameters between 2.5 and 10 μm). FB exhibits the higher proportion of di-n-octyl phthalate (DnOP) in PM<sub>2.5-10</sub> than PM<sub>2.5</sub>. RT shows a distinguishable profile with high abundances of rubber. The abundance of 2-benzothiazolyl-Nmorpholinosulfide (OBS) in PM<sub>2.5-10</sub> was twice that in PM<sub>2.5</sub> for RT. Polystyrene (PS) is the most abundant MP in AF. LB shows the distinguishing feature of benzothiazoles, especially OBS and Ncyclohexyl-2-benzothiazolesulfenamide (CBS). The eco-health risk assessments reveal combustion-derived MPs (Plastic Burning and Fruit-bag Burning) indicated the highest ecological risk (Level III). Elevated hazard indices to human health were observed in LB and PB, primarily attributed to bis(2-ethylhexyl) phthalate (DEHP). Notably, PMMA, polyethylene terephthalate (PET), polyethylene (PE), bisphenol A (BPA), and phthalates (PAEs) emerged as key drivers of oxidative stress of PMs. This study advances the understanding of atmospheric MPs, offering critical insights for source tracking and risk assessment to mitigate their eco-health effects.

**Keywords:** Microplastic and plasticizer emission source; size distribution; phthalates (PAEs);

eco-health risk; reactive oxygen species (ROS)

#### 1. Introduction

Global plastic production has increased exponentially after 1990, resulting in serious environmental contamination (Geyer et al., 2017; Klein et al., 2023). Waste plastics have accumulated in the environment to be degraded into plastic debris under the influences of UVradiation, mechanic abrasion and temperature changes (Peeken et al., 2018; Akhbarizadeh et al., 2021). Microplastics (MPs) are plastic particles with 5 mm-1 µm in size (Can-Guven, 2021). Research on MPs pollution initially focused on aquatic and terrestrial ecosystems, but recent years have seen growing attention to atmospheric MP pollution (Allen et al., 2020). Understanding the sources of atmospheric MPs can assist develop efficient MPs management strategy. Common atmospheric MPs sources include waste incineration, agricultural activities, and road traffic (Panko et al., 2013; Luo et al., 2022; Yang et al., 2024; Chen et al., 2024). Incineration activities can lead to the fragmentation of plastics, accelerating the release of MPs (Luo et al., 2022; Luo et al., 2024b). Yang et al. (2021) have estimated that per metric ton of plastic can potentially produce 360 to 102,000 MPs, primarily composed of polypropylene (PP) and polystyrene (PS). In addition to industrial incineration processes, open burning activities in daily life also contribute significantly to atmospheric MPs. Due to the relatively limited facilities and means of waste disposal, residents in rural areas often resort to open burning when disposing of plastic waste (Pathak et al., 2023). In addition, given the flammability of plastics, residents also tend to use plastics as igniters or even burn them directly when using stoves for cooking or heating, which is an important household source of MPs. Agricultural activities also a significant contributor to atmospheric MPs (Jin et al., 2022; Yuan et al., 2025). The large consumption of plastic film combined with short life

cycle result in a number of films being left in farming soil, then transforming into MPs via degradation or fragmentation (Brahney et al., 2021; Wang et al., 2022a; Aini et al., 2023). Agricultural activities (e.g., plowing and harvesting), by increasing soil disturbance, may cause the resuspension of MPs into the atmosphere (Jin et al., 2022; Lakhiar et al., 2024). Furthermore, tire and road wear microplastics (TRWMPs), producing from the interaction between tires and the road surface, is a significant source of atmospheric MPs (Panko et al., 2013; Liu et al., 2023; Xu et al., 2024b). Evangeliou et al. (2020) have estimated that annual total global tire wear particle emissions were 2907 kt y<sup>-1</sup>, with 29 and 288 kt y<sup>-1</sup> for PM<sub>2.5</sub> (particulate matter with aerodynamic diameters ≤ 2.5  $\mu$ m) and PM<sub>10</sub> ( $\leq$  10  $\mu$ m), respectively. Liu et al. (2023) have showed rubbers were the dominant compounds of TRWMPs in PM2.5 in tunnels, including natural rubber (NR), styrene-butadiene rubber (SBR), and butadiene rubber (BR) polymers. Current research on atmospheric MPs sources focuses on industrial emissions and natural processes, but neglects air pollution sources closely related to daily life sources. Given that living sources significantly affect human health, this study pays particular attention to such sources.

Plasticizers are widely used in the production of plastics in order to achieve the desired material properties (Demir and Ulutan, 2013). Since plasticizers are not chemically bound to the plastic products, they can easily diffuse into the surrounding environment during the life-time (Demir and Ulutan, 2013; Yadav et al., 2017). Phthalate esters (PAEs), Benzothiazoles (BTs), and Bisphenol A (BPA) are the most common plastic additives that are ubiquitous in the environment and pose potential health risks. PAEs are the most widely used plasticizers globally, dominating the plastic additive market. He et al. (2020) demonstrated that during 2007-2017, the annual global production

of PAEs increased from 2.7 million tons to 6 million tons. China is recognized as the largest importer of PAEs worldwide (Cui et al., 2025). BTs are extensively used in automotive tires and agrochemicals. High concentrations of BTs were discovered in the street runoff, suggesting that these tire material-related compounds can persevere in the environment (Zhang et al., 2018). Exposure to BTs may result in central nervous system depression, liver and kidney damage, dermatitis, and pulmonary irritation (Ginsberg et al., 2011). BPA as a common industrial chemical component in many products, has steadily grown over the last 50 years (Corrales et al., 2015). Growth of global production has consistently ranged between 0% and 5% annually (Corrales et al., 2015). PAEs and BPA considered as endocrine disruptors, have been demonstrated to impair reproductive function and development in laboratory animals (Wang et al., 2019).

Previous studies have investigated the emission characteristics of plasticizers from various sources. Simoneit et al. (2005) illustrated that the major plasticizers detected in PMs from open-burning of plastics were dibutyl phthalate (DBP), diethylhexyl adipate (DEHA), and diethylhexyl phthalate (DEHP). Zeng et al. (2020) reported phthalate concentrations in greenhouses air were higher than that in ambient air. Liu et al. (2023) found that phthalates were the most dominant plasticizer compositions in tunnel PM<sub>2.5</sub>, accounting for 64.8% of the detected plasticizers. Zhang et al. (2018) demonstrated that tire material-related compounds, benzothiazole (BT) and 2-hydroxybenzothiazole (2-OH-BT) were the major compounds in both tire and road dust samples. The majority of existing studies on atmospheric MPs and plasticizers have focused on analyzing the emission characteristics of individual source and lacked a comprehensive and comparative analysis of the MPs emission profiles of various sources.

MPs and plasticizers can remain suspended and spread to other areas when they emitted from the sources into the air (Gasperi et al., 2018). Airborne MPs can easily enter the human body directly via respiration compared to other environmental exposure pathways, posing a serious health concern (Liao et al., 2021; Luo and Guo, 2025). Recent studies suggest that these inhaled pollutants can promote reactive oxygen species (ROS) generation (Wang et al., 2024). Oxidative potential is a metric reflecting the ability of inhaled pollutants to produce ROS, serving as a critical indicator of PM toxicity (Jiang et al., 2019; Bates et al., 2019; Luo et al., 2024c). ROS overproduction acts as a central driver of oxidative stress, which can damage biomolecules and disrupt cellular functions (Bates et al., 2019; Jiang et al., 2019). Previous studies have demonstrated that metals and organic compounds can affect the oxidative potential of PMs (Ghanem et al., 2021; Luo et al., 2023). However, most of the studies on MPs and plasticizers have focused on their environmental occurrence rather than systematic health risk assessments from atmospheric pollution sources.

The Guanzhong Plain located in the central of Shaanxi Province, northwestern China, inevitably consumes a large number of plastics with a developed economy and a large population (Chen et al., 2022; Wang et al., 2022b; Xu et al., 2024a). The environmental conditions of strong wind and ultraviolet ray in this area exacerbate the problem of atmospheric MPs pollution (Liu et al., 2017). There is a notable absence of systematic comparative analyses examining the emission profiles across various emission sources, which is the key to controlling MPs pollution. The aims of this research are to (i) characterize the distributions of MPs and plasticizers in dual-size PMs (PM<sub>2.5</sub>, PM<sub>2.5-10</sub>) from typical MP sources (anthropogenic sources from daily life) in the Guanzhong Plain, (ii) obtain MPs and plasticizers tracers for the five typical MP sources, and (iii) evaluate the

health risks of MPs and plasticizers in  $PM_{2.5}$  and  $PM_{10}$ . This study could provide valuable scientific support for the development of targeted pollution control strategies, as well as sustainable improvement of the regional environment and public health protection.

## 2. Methods

## 2.1 Sample collection and gravimetric method

During January and February 2024, PM<sub>2.5</sub> and PM<sub>10</sub> samples were collected simultaneously from five distinct sources in three key cities of the Guanzhong Plain: Xi'an, Tongchuan, and Xianyang (Figure S1). The selected sources included Plastic Burning, Fruit-bag Burning, Road Traffic, Agricultural Film, and Livestock Breeding. It should be noted that plastics of Plastic Burning source incineration including plastic bags, bottles, disposable tableware, foam boxes, and other plastic daily necessities. Fruit bags are typically lightweight, thin-film, which are designed for single-use and are often discarded after a short period, differing from common household plastic waste. These bags are usually made from low-density polyethylene and Nylon, which is known for its flexibility and transparency(Ali et al., 2021; Yang et al., 2022). The Guanzhong Plain is an important fruit production base in China, with the highest consumption of fruit bags. Local residents often use the above-mentioned plastic products to ignite solid fuels for indoor heating or cooking. The reason we distinguish between Plastic Burning and Fruit-bag Burning rather than classifying them as a single combustion source is that the wax layer in fruit bags cannot be separated from the plastic. This is a featured source in the Guanzhong region (This is also quite common in fruit producing areas in northern China), and local residents typically burn fruit bags directly without separating the wax. Table 1 provides a summary of the essential details for each source.

M

147 h

iii

e

o

1

a

a

w

155 s

The specimens were gathered using pre-fired quartz-fiber filters (QM/A, PALL, Ann Arbor, MI, USA) with a diameter of 47 mm, which had been subjected to a temperature of 800 °C for 3 hours (Wang et al., 2022b). MiniVOL samplers (Airmetrics, Springfield, OR, USA), which are inertial impactors, were employed for collection, operating at a steady flow rate of 5 L min<sup>-1</sup> (Wang et al., 2022b) (Figure S1). Sampling periods for each source ranged from 2 to 24 hours, depending on the emission amount. In Agricultural Film and Livestock Breeding, the sampler was set at about 1.5 m height, corresponding to the human breathing height. For Plastic Burning, Fruit-bag Burning, and Road Traffic sources, the sampling heights were related to the height of the chimney and flyover, about 3-4 m above the ground. The field blank of each type of source was synchronously collected with active sampling. Unused filters (the same batch as sampling filters) were loaded into identical sampling devices, which were placed adjacent to operational samplers for the entire duration of one sampling event.

Table 1 Basic sampling information of target emission sources

| Emission             | Sampling     | Sampling               | Sample | Sampling location                                                           |
|----------------------|--------------|------------------------|--------|-----------------------------------------------------------------------------|
| source               | duration (h) | height                 | No.    |                                                                             |
| Plastic Burning      | 2.0          | 3-4 m above the ground | 5      | Open space, about 1 m downwind of chimney of rural household stove in rural |
| Fruit-bag<br>Burning | 2.0          | 3-4 m above the ground | 5      | Xianyang                                                                    |
| Road Traffic         | 13.6-14.1    | 3 m above the ground   | 5      | Open space, flyovers on traffic arteries in downtown Xi'an                  |
| Agricultural         | 24           | 1.5 m                  | 5      | Open space, about 2 m away                                                  |
| Film                 |              | above the ground       |        | from the greenhouse in farmland in rural Tongchuan                          |
| Livestock            | 2.5-3.5      | 1.5 m                  | 5      | About 1 m from the feed trough                                              |
| Breeding             |              | above the              |        | in a cow shed of approximately 8                                            |

Filters were transferred using stainless steel tweezers into pre-labeled clean glass cassettes after collection and frozen at -20 °C until chemical analyses. An electronic microbalance (± 1 µg sensitivity, ME 5-F, Sartorius, Germany), with an anti-static instrument, was used to weigh the filters before and after sampling (Wang et al., 2022b). Cotton lab coats and nitrile gloves were utilized during sampling, while the use of plastic materials was minimized (Bogdanowicz et al., 2021).

## 2.2 Chemical analysis

This study quantified eight kinds of microplastics and three classes of plasticizers (phthalates, benzothiazole and its derivatives, bisphenol A) in  $PM_{2.5}$  and  $PM_{10}$  samples.

## 2.2.1 Microplastics (MPs)

To quantify the contents of the MPs, a setup was employed where a Curie-point pyrolyzer (JHS-3, Japan Analytical Industry Co., Ltd) was connected to a gas chromatography-mass spectrometry (GC/MS) system (7890GC/5975MS, Agilent Technology, USA) (Liu et al., 2023). The pyrolysates of polyethylene (PE), polypropylene (PP), polystyrene (PS), polyethylene terephthalate (PET), poly(methyl methacrylate) (PMMA), natural rubber (NR), styrene-butadiene rubber (SBR), and butadiene rubber (BR) were identified using mass spectrum fragments, retention times, and target product intensities compared to plastic standards. All standards excepting rubbers (99%, JSR Corporation) were purchased from Dupont (≥ 98%, USA). The quantified markers for the pyrolyzed compounds are shown in Table S1. Details regarding preparation of samples, instrument configurations (Sun et al., 2022) are available in S1...

# 2.2.2 Phthalates (PAEs)

An in-port thermal tube (78 mm long, 4 mm I.D., 6.35 mm O.D., Agilent Technology, USA) coupled with a GC/MS system (7890GC/5975MS, Agilent Technology, USA) was utilized to analyze phthalates, including dimethylphthalate (DMP), diethyl phthalate (DEP), di-n-butyl phthalate (DBP), butyl benzyl phthalate (BBP), bis(2-ethylhexyl)phthalate (DEHP), and di-n-octyl phthalate (DnOP). All PAEs were purchased from Sigma-Aldrich (≥98%, Steinheim, Germany). Aliquots of the filters (1.578 cm²) were diced into smaller fragments, augmented with the internal standard Chrysene-d12 (96%, LGC Standard Limited, United States), and then inserted into thermal tubes (78 mm long, 4 mm I.D., 6.35 mm O.D., Agilent Technology, USA) for analyses (Liu et al., 2023). The sample tube was inserted directly into a GC injection port set to an initial temperature of 50 °C (Wang et al., 2016). Detailed sample preparation procedures and analytical procedures (Ho et al., 2019; Liu et al., 2023) are provided in S2.

# 2.2.3 Benzothiazole and its derivatives (BTs)

Nine types of benzothiazole (97%, Thermofisher Scientific Co., LTD, Waltham, MA, United States) related compounds were quantified, including benzothiazole (BT), 2-hydroxy benzothiazole (HOBT), 2-mercaptobenzothiazole (MBT), 2-aminobenzothiazole (2-NH2-BT), 2-(methylthio)benzothiazole (MTBT), 2-(4-morpholinyl)benzothiazole (24MoBT), N-cyclohexyl-2-benzothiazolamine (NCBA), 2-benzothiazolyl-N-morpholinosulfide (OBS), N-cyclohexyl-2-benzothiazolesulfenamide (CBS). The appropriate filter sample was cut (1.578 cm²) and spiked with an IS of benzothiazole-d4 (95%, LGC Standard Limited, United States). After a series of extraction and concentration procedures (S2), the target analytes were washed out with 5 mL of methanol (HPLC grade, Fisher Chemical, USA). Before the analysis, the eluates were then dried to 1 mL

under a stream of nitrogen (Zhang et al., 2018). Target analytes were separated using an ultraperformance liquid chromatography system (UPLC; ACQUITY, Waters, USA) and subsequently identified with a triple quadrupole mass spectrometer (ESI-MS/MS; Xevo TQ-S, Waters, USA). Analytical details (Zhang et al., 2018) are provided in S2.

## 2.2.4 Bisphenol A (BPA)

Quantification of total BPA and separation from the matrix components were carried out by LC-fluorescence detection (García-Prieto et al., 2008). Mobile phase was composed of acetonitrile and water (Jian-Ke et al., 2011). The BPA standard was obtained from Sigma-Aldrich (USA). Moreover, all employed solvents and diluents were of HPLC grade and filtered through 0.45 µm membranes. The sample extracted was separated by a PerkinElmer Brownlee<sup>TM</sup> HRes Biphenyl 1.9 µm, 50 × 2.1 mm column with isocratic elution program of water: acetonitrile (6:4) at 0.5 mL min<sup>-1</sup> for 4 min. The target analyte was measured using a fluorescence detector at excitation and emission wavelengths of 275 nm and 313 nm, respectively (García-Prieto et al., 2008). BPA levels were quantified based on measured peak areas (García-Prieto et al., 2008).

# 2.3 Quality assurance/Quality control (QA/QC)

The flow rates of all samplers were calibrated using a mass flowmeter (Model 4140, TSI, Shoreview, MN, USA) before and after each sampling cycle. All quartz filters used in this study were preheated at 800°C for 3 h to remove any potential contaminants and then cooled before use. To minimize experimental error, sampling was conducted in duplicate for each particle size of each source. For the chemical measurements, one in every 10 samples was reanalyzed for quantity assurance purposes, and the standard deviation errors of replicate trials were within 10% for the

pyrolysis analyses. Calibration curves were established using reference standards. The linearities of the standard calibration curves were > 0.987. The standard deviations of the pyrolyzed standard were within 94.1% to 98.3%. Background contamination (Table S3) was monitored by processing operational blanks (unexposed filters) simultaneously with field samples.

2.4 Oxidative potential determination with DTT assay

Four 0.526 cm<sup>2</sup> punches per sample from different sources were individually dissolved in 5 mL methanol (HPLC grade, Fisher Chemical, USA) in an amber centrifuge tube and ultrasonically extracted for 2 h. The PM extract was used for the subsequent analysis.

The Dithiothreitol (DTT) consumption in this study was quantified following the methodology established by Luo et al. (2024c). 4 mL of sample extract was combined with 1 mL of 1 mM DTT solution ( $\geq$ 98%, Meryer; pH 7.4 buffer), yielding a final concentration of 200  $\mu$ M. At each time point (0, 5, 15, 30, 45, and 60 min), 0.5 mL of the DTT reaction mixture was added to the amber centrifuge tube preloaded with 0.5 mL of trichloroacetic acid (1%, w/v) to terminate the reaction. Subsequently, 25  $\mu$ L of 10 mM 5,5°-dithiobis-(2-nitrobenzoic acid) (DTNB,  $\geq$ 98%, Meryer) and 1 mL of 1 M Tris-HCl buffer were added to each tube. The solutions (200  $\mu$ L) were transferred to 96-well plates, and absorbance was measured at 412 nm using a microplate reader (Flex Station 3 Multi-Mode, Molecular Devices). The volume-normalized DTT consumption rates for each sample were calculated from absorbance measurements taken at predetermined time points (nmol min<sup>-1</sup> m<sup>-3</sup>).

To ensure the accuracy of the results, the entire experiment was performed under dark conditions. Prior to sample analysis, a standard curve was generated by measuring the absorbance

of 11 DTT concentration gradients within the range of 0 to 450 µmol L<sup>-1</sup>, achieving a correlation coefficient (R<sup>2</sup>) of 0.9997. Pure methanol solution was used as a blank control, which was processed and measured in the same manner as the samples. The DTT consumption rate of each sample was corrected using the DTT consumption rate of the blank. Each batch of samples and methanol blanks was measured in duplicate to verify experimental reproducibility. The linear fitting R<sup>2</sup> for DTT consumption rates was consistently greater than 0.9, and the coefficient of variation (standard deviation) for parallel experiments was less than 15%.

#### 2.5 Risk assessment model

To evaluate the potential ecological risks, the hazard indices of various MPs were estimated using the Formula 1 (Xu et al., 2018; Wang et al., 2021a). The risk index (H) was calculated by multiplying the proportion  $(P_n)$  of each polymer identified in MPs by its respective hazard score  $(S_n)$  (Lithner et al., 2011).

$$H = \sum Pn \times Sn \quad (1)$$

The average daily exposure dose (ADD) via respiratory inhalation was calculated by Formula 2, defined by the U.S. Environmental Protection Agency (U.S.Epa, 1989; Liu et al., 2023). The non-carcinogenic and carcinogenic health risks of MPs and plasticizers were quantified using the hazard quotient (HQ) (Formula 3) and incremental lifetime cancer risk (ILCR) (Formula 4), respectively.

$$ADD = \frac{C \times ET \times IR \times EF \times ED}{AT \times BW}$$
 (2)
$$HQ = \frac{ADD}{RfD}$$
 (3)

$$ILCR = ADD \times SF (4)$$

where C represents the measured mass concentration of MPs and plasticizers from five sources.

Exposure parameters included: ET (exposure time, 0.5 h d<sup>-1</sup> for combustion sources (Plastic Burning and Fruit-bag Burning), 1.5 h d<sup>-1</sup> for others), IR (inhalation rate, 20 m<sup>3</sup> d<sup>-1</sup>), EF (exposure frequency, 120 d y<sup>-1</sup> for Plastic Burning and Fruit-bag Burning, 350 d y<sup>-1</sup> for others), ED (exposure duration, 30 years), AT (average exposure time, ED×365 d y<sup>-1</sup>×24 h), and BW (adult body weight, 70 kg)(Liu et al., 2023). Reference dose (RfD) and slope factor (SF) were obtained from the Integrated Risk Information System of U.S. EPA (https://www.epa.gov/iris) and Ma et al. (2020) as detailed in Table S7.

2.6 Data analysis and statistical method

Data entry and organization were conducted using Excel 2016 (Microsoft Corporation, Redmond, WA, USA), while one-way analysis of variance (ANOVA) was performed with SPSS 26.0 (IBM, Armonk, NY, USA). Spearman correlation analysis was used to assess the relationships of MPs and plasticizers with ROS, respectively. Additionally, all data are presented as mean ± standard deviation, with significant differences denoted by P<0.05.

The Source-Pathway-Receptor (SPR) model serves as a key tool for illustrating how environmental pollutants travel from their origins, navigate various pathways, and ultimately reach potential receptors (Waldschläger et al., 2020).

- 3. Results and discussion
- 3.1 Concentrations of microplastic and plasticizer
  - For the convenience of comparison, we subtracted the concentrations of MPs and plasticizers in  $PM_{2.5}$  from  $PM_{10}$  in this study to obtain their concentrations in coarser particulate matter ( $PM_{2.5}$ 10). The total concentrations of MPs and plasticizers in  $PM_{2.5}$  and  $PM_{2.5-10}$  from five different sources

are presented in Figure 1. MPs were more enriched in PM<sub>2.5-10</sub> in Fruit-bag Burning source (59% of PM<sub>10</sub>), while higher in PM<sub>2.5</sub> for the remaining four sources (Plastic Burning, Road Traffic, Agricultural Film, and Livestock Breeding). The fruit bags are coated with wax layer for enhancing the waterproofing and durability of the material. The presence of this wax layer may affect particle formation during combustion, contributing to the creation of larger agglomerates and thus a higher proportion of coarse particles. Notably, MPs in Plastic Burning and Livestock Breeding constituted a comparable proportion in both fine and coarse fractions, both close to 50%. The variable order of MPs concentrations in the five sources in PM<sub>2.5-10</sub> was roughly consistent with that of PM<sub>2.5</sub>, showing Plastic Burning > Fruit-bag Burning > Livestock Breeding > Agricultural Film > Road Traffic. The average concentration of MPs was ranging from  $77.7 \pm 25.3$  (Road Traffic) to  $1906 \pm$ 587 (Plastic Burning) ng m<sup>-3</sup> in fine fraction and from  $41.5 \pm 11.7$  (Road Traffic) to  $1634 \pm 20.3$ (Plastic Burning) ng m<sup>-3</sup> in coarse fraction of PMs, as summarized in Table S2. The highest MPs concentrations in fine and coarse PMs in Plastic Burning source were both ~5 times higher than the averages of that in other sources. One possible explanation for this is that plastic waste can be fragmented into MPs during the process of combustion (Yang et al., 2021; Luo et al., 2024a). Another important pathway for elevated MPs from Plastic Burning source is the resuspension of bottom ash (Yang et al., 2021).

**Figure 1** Average concentrations of MPs (a) and plasticizers (b) in PM<sub>2.5</sub> and PM<sub>2.5-10</sub> from five sources (PB: Plastic Burning, FB: Fruit-bag Burning, RT: Road Traffic, AF: Agricultural Film, LB: Livestock Breeding).

The total concentrations of the plasticizers in the samples were one order of magnitude higher than those of MPs (Table S2). The mass concentrations of plasticizers were higher in PM<sub>2.5</sub> than in PM<sub>2.5-10</sub> for Fruit-bag Burning, Road Traffic, Agricultural Film, and Livestock Breeding sources, especially with the value of 80% in fine particles from Road Traffic. Both MPs and plasticizers in Road Traffic were more abundant in PM<sub>2.5</sub>, which enhanced the potential for long-range transport and respiratory penetration. Therefore, even though the emission concentrations from Road Traffic source were lower, the potential environmental and health risks posed by road traffic cannot be overlooked. Conversely, plasticizers in Plastic Burning source were abundant in PM<sub>2.5-10</sub> (59%). The highest concentration values of plasticizer in this study were also observed in Plastic Burning (15.6 ± 5.61  $\mu$ g m<sup>-3</sup> in PM<sub>2.5</sub>, 22.3 ± 1.68  $\mu$ g m<sup>-3</sup> in PM<sub>2.5-10</sub>), followed by Fruit-bag Burning (4.53 ± 0.39  $\mu$ g m<sup>-3</sup> in PM<sub>2.5</sub>, 2.75 ± 0.65  $\mu$ g m<sup>-3</sup> in PM<sub>2.5-10</sub>). This is because plastic products contain many additives to enhance their performance (Do et al., 2022). Many additives are not covalently bound to the polymer matrix, resulting in the liberation of plastic additives during the crushing and

combustion (Do et al., 2022; Billings et al., 2023). Furthermore, Livestock Breeding exhibited a higher emission for plasticizers in non-combustion sources (Road Traffic and Agricultural Film; P<0.05 in  $PM_{2.5}$ ), with the values of  $2.17\pm1.05$  and  $1.16\pm0.88~\mu g$  m<sup>-3</sup> respectively for  $PM_{2.5}$  and  $PM_{2.5-10}$ . The lack of an effective plastic recycling and disposal system under the traditional retail farming may exacerbate the release of plasticizers.

## 3.2 Chemical composition of microplastics

The proportions of MPs identified in PM<sub>2.5</sub> and PM<sub>2.5-10</sub> for the five sources are presented in Figure 2. The composition of MPs from five sources varied greatly, but no significant size distribution difference in the same source. Road Traffic exhibited distinctive features from other four rural sources with the high proportions of both BR+SBR and NR in PM<sub>2.5</sub> (46.2  $\pm$  3.31% and 33.3  $\pm$  2.65% of MPs, respectively) and PM<sub>2.5-10</sub> (50.7%  $\pm$  2.94 and 18.6%  $\pm$  0.79 of MPs, respectively), which are the basic material of tire treads. In previous studies, BR+SBR was observed to be the predominate MPs in light-duty vehicle tires in the tunnel PM<sub>2.5</sub>, and conversely, NR is extensively used in tire treads for trucks (Liu et al., 2023). The Road Traffic sample collection in this work was done in the downtown flyover in urban Xi'an, where light-duty cars are the dominant vehicle type, explaining the high proportion of BR+SBR than NR both in PM<sub>2.5</sub> and PM<sub>2.5-10</sub>.

**Figure 2** Chemical composition of microplastics in PM<sub>2.5-10</sub> and PM<sub>2.5</sub> from the five sources (PB: Plastic Burning, FB: Fruit-bag Burning, RT: Road Traffic, AF: Agricultural Film, LB: Livestock Breeding).

The MP compositions of Plastic Burning, Fruit-bag Burning, Agricultural Film, and Livestock Breeding were relatively similar. PET was the most common polymer type in MPs (Figure 2), which is widely used in the production of textiles. Plastic Burning source inevitably included a certain amount of waste textiles, inducing the release of PET (Yang et al., 2021). Moreover, PET is widely applied in packaging and agriculture due to its advantageous properties, such as good strength, durability, elasticity, clarity, etc. (Liu et al., 2019; Lu et al., 2024). These materials may break into MPs due to wear and tear, subsequently discharging into the agricultural and breeding environment. Moreover, the highest proportion of PS was found in Agricultural Film source. Agricultural facilities made of PS (e.g., lamp-chimneys, electrical devices) in greenhouses may influence the MPs composition of Agricultural Film source (Qi et al., 2023).

## 3.3 Chemical composition of plasticizers

PAEs were the most prevalent (> 90%) among the three plasticizers in the five sources in this study. PAEs have been the most widely used plasticizer and the global production is expected to reach 500 million tons by 2050 (Huang et al., 2021; Billings et al., 2024). The levels of the total PAEs ranged from  $468 \pm 175$  ng m<sup>-3</sup> (Agricultural Film)- $15640 \pm 5609$  ng m<sup>-3</sup> (Plastic Burning) in  $PM_{2.5}$  and  $115 \pm 54.4$  ng m<sup>-3</sup> (Road Traffic)-22274  $\pm 1680$  ng m<sup>-3</sup> (Plastic Burning) in  $PM_{2.5-10}$  (Table S2). The percentages of BTs and BPA among the three detected plasticizer types were below 2%. The highest concentrations of PAEs, BTs, and BPA still appeared in PB among five sources. The concentrations of BPA in Plastic Burning and Fruit-bag Burning sources were an order of magnitude higher than Road Traffic and Agricultural Film (Table S2). The result indicated that Plastic Burning is the primary emission source of atmospheric plasticizers, in agreement with prior research (Zhen et al., 2019; Chandra and Chakraborty, 2023). Compared to other sources, Road Traffic source demonstrated a higher concentration of BTs  $(34.8 \pm 13.0 \text{ ng m}^{-3} \text{ for PM}_{2.5}, P<0.05; 12.9 \pm 7.28 \text{ ng m}^{-3} \text{ for PM}_{2.5-10})$  (Table S2). This may be related to the widespread use of BTs in tire manufacturing and these additives are released into the

air during friction between tire and road surface (Liu et al., 2023). At the same time, some tire rubber substances were also involved in the Plastic Burning source of this study. Livestock Breeding exhibited the highest emission of BPA among non-combustion sources, with the values of  $50.9 \pm 27.1$  and  $38.6 \pm 22.2$  ng m<sup>-3</sup> respectively for PM<sub>2.5</sub> and PM<sub>2.5-10</sub>, higher than Road Traffic (4.43  $\pm 1.45$  and  $7.8 \pm 0.9$  ng m<sup>-3</sup>, P<0.05) and Agricultural Film (1.4  $\pm 0.71$  and 4.29  $\pm 6.68$  ng m<sup>-3</sup>, P<0.05), partly due to the migration of BPA from animal feed plastic packaging into the air (Wang et al., 2021c; Wang et al., 2021b). Furthermore, BTs, PAEs, and BPA from sources except for Plastic

Burning were prevalent in PM<sub>2.5</sub> relative to PM<sub>2.5-10</sub>, contrary to the results reported by Nunez et al. (2020). This discrepancy may be attributed to differences in pollution sources. Nunez et al. (2020) demonstrated that port industrial activities (e.g., cargo handling and industrial emissions) predominantly generated coarse PMs, resulting in higher concentrations of plasticizers in this fraction. In contrast, high temperature in Plastic Burning source promoted the formation of fine particles with larger surface area enhancing the adsorption of plasticizers.

## 3.3.1 Compositions and distributions of PAEs

DnOP was the most abundant PAE specie across Plastic Burning, Fruit-bag Burning, Road Traffic, and Agricultural Film. For Fruit-bag Burning source, DnOP was significantly more prevalent in PM<sub>2,5-10</sub>, accounting for  $51 \pm 12\%$  of the total PAEs, compared to  $36 \pm 1.8\%$  in PM<sub>2,5</sub>. Conversely, DnOP was more abundant in fine  $(59 \pm 1.0\%)$  fraction of PMs than coarse  $(44 \pm 0.4\%)$ in Road Traffic. As a common plasticizer, DnOP possesses a high molecular weight and low volatility, increasing its persistence in the environment. In addition to DnOP, DEHP, and BBP were also identified as the major components in five sources. DEHP was a second abundant PAE component in Road Traffic (23  $\pm$  0.5% and 30  $\pm$  0.2% of PAEs in PM<sub>2.5</sub> and PM<sub>2.5-10</sub>), as it has a high consumption in plasticizers market, especially in automobile industry (Zhen et al., 2019). While the lowest percentage of DEHP in Agricultural Film in both  $PM_{2.5}$  and  $PM_{2.5-10}$  (13  $\pm$  0.1%,  $12 \pm 0.3\%$ ) among five sources is the significant characteristic for Agricultural Film. BBP was the most abundant PAE in PM<sub>2.5-10</sub> in Livestock Breeding, and the proportion was higher in coarse (40  $\pm$  14%) than fine (28  $\pm$  1.6%) PMs. Moreover, as shown in Figure 3, the proportions of sum of DMP, DEP, and DBP were below 30% in both PM<sub>2.5</sub> and PM<sub>2.5-10</sub>, and were even below 15% in Fruit-bag

Burning and Road Traffic. The proportion of DEP ( $12 \pm 5.1\%$  and  $15 \pm 0.4\%$  in PM<sub>2.5</sub> and PM<sub>2.5-10</sub>, respectively) was the highest in Plastic Burning compared to other sources, which could be used as the source marker (Figure 3a).

Figure 3 Mass proportions of phthalates (PAEs) (a, b, c, d, e) and benzothiazole and its

derivatives (BTs) (f, g, h, i, j) in PM<sub>2.5-10</sub> (outer ring) and PM<sub>2.5</sub> (inner ring) in the five typical sources (PB: Plastic Burning, FB: Fruit-bag Burning, RT: Road Traffic, AF: Agricultural Film, LB: Livestock Breeding).

## 3.3.2 Compositions and distributions of BTs

The distribution patterns of BTs in the five typical MP sources in PM<sub>2.5</sub> and PM<sub>2.5-10</sub> were more different than PAEs. The compositions of Plastic Burning and Agricultural Film were quite similar, may proving that rural households use discarded agricultural film for heating or cooking during indoor fuel combustion. HOBT was the most abundant compound in Plastic Burning and Agricultural Film, with values of  $42 \pm 1.5\%$ ,  $31 \pm 0.0\%$ , respectively for PM<sub>2.5</sub> and  $28 \pm 2.5\%$ , 35 ± 5.2% for PM<sub>2.5-10</sub> (Figure 3f, 3i). Furthermore, MBT was more prominent than other species for Fruit-bag Burning, with the values of more than 20%. The abundances of OBS in PM<sub>2.5-10</sub> (36  $\pm$ 0.1%) were higher than that in PM<sub>2.5</sub> (18  $\pm$  1.8%) for Road Traffic. Some previous studies have implied the main use of OBS in tire manufacture (Liao et al., 2018; Liu et al., 2023). BT in Road Traffic was more predominant in PM<sub>2.5</sub> (27  $\pm$  1.0%) compared to PM<sub>2.5-10</sub> (10  $\pm$  1.0%), aligning with the prevalence of MPs and plasticizers in PM<sub>2.5</sub> in Road Traffic. A high concentration of BT in tire debris was reported from Sweden demonstrating that tire wear is the main cause of road traffic pollution (Avagyan et al., 2014). OBS+CBS accounted for more than 70% of BTs only in Livestock Breeding source, which were significantly higher than those in other sources and could be used as the source markers.

3.4 Source profiles of MPs and plasticizers

The source profiles of MPs, BTs, PAEs, and BPA in PM<sub>10</sub> and PM<sub>2.5</sub> emitted from the five

emission sources are shown in Figure 4. The distribution patterns of each chemical species exhibited insignificant differences between PM<sub>2.5</sub> and PM<sub>10</sub>. DnOP emerged as the predominant contributor across all sources, with Plastic Burning being the most significant, representing  $1.4 \pm 0.33\%$  and  $2.9 \pm 0.06\%$  of PM<sub>2.5</sub> and PM<sub>10</sub> mass concentrations, respectively. The profiles of the combustion sources (Plastic Burning and Fruit-bag Burning) were more similar. However, PMMA exhibited a higher proportion in Plastic Burning (0.085  $\pm$  0.033% and 0.23  $\pm$  0.01% in PM<sub>2.5</sub> and PM<sub>10</sub>, respectively) compared to Fruit-bag Burning (0.023  $\pm$  0.001% and 0.041  $\pm$  0.004%). In addition, HOBT, the most abundant BT derivative in the current study, accounted for  $0.024 \pm 0.015\%$  in PM<sub>2.5</sub> and  $0.037 \pm 0.003\%$  in PM<sub>10</sub> in Plastic Burning, but less than 0.001% in Fruit-bag Burning. For non-combustion sources, Road Traffic was significantly influenced by tire wear particles, which characterized by high abundances of tire-related material, such as NR (0.047  $\pm$  0.005%) and BR+SBR (0.072  $\pm$  0.008%). Liu et al. (2023) revealed that NR and other rubber particles were emitted at high levels in tunnel traffic, emerging as the dominant microplastic in traffic-dominated environments. Moreover, 24MoBT constituted the highest percentage of BTs in Road Traffic source, which can also be used as the indicator of vehicle emission plasticizer. PS constituted a higher proportion in PMs than PET in Agricultural Film, with  $0.18 \pm 0.04\%$ ,  $0.14 \pm 0.02\%$  respectively to  $PM_{2.5}$  and  $0.16 \pm 0.19\%$ ,  $0.15 \pm 0.15\%$  to  $PM_{10}$ . This is in line with the findings of Liu et al. (2019), who documented that PS were the predominant. OBS and CBS were the prevalent BT compounds in Livestock Breeding, and BPA played a more significant contribution to PM<sub>10</sub> than PM<sub>2.5</sub>.

Figure 4 Source profiles of microplastics and plasticizers in PM<sub>2.5</sub> and PM<sub>10</sub> (The black arrows

indicate the source markers).

## 3.5 Eco-health significance

In this section, a comprehensive eco-health risk evaluation system was established to provide scientific support for estimating the hazards of MP and plasticizers from different sources. 1) The transport pathways of MPs and plasticizers from Plastic Burning, Road Traffic, and Agricultural Film sources were analyzed to clarify the exposure routes from "source" to "receptor" (Figure 5).

2) The ecological and health risks of MPs and plasticizers were assessed through different evaluation metrics (H, HI, ILCR, and oxidation potential).

## 3.5.1 Transport pathways of MPs and plasticizers

As shown in Figure 5, plastic combustion emitted MPs and attached plasticizers into the ambient air (Velis and Cook, 2021); the residual in the bottom ash can break into MPs via wind abrasion, then re-suspending into the air or depositing onto surrounding soil or into water with a risk of entering the food chain (Yang et al., 2021; Velis and Cook, 2021; Pathak et al., 2024). Small microplastics (micro-rubber) from Road Traffic emitted as airborne fine particles or trapped in the road surface, which can enter the water by surface runoff, migrating and transforming in different environmental media (Kole et al., 2017). Under ultraviolet degradation and wind erosion, agricultural films can release MPs and plasticizers into the air directly, while larger particles were deposited in farmland (Song et al., 2017). Disturbed by agricultural activities and wind, MPs created by residual films in the soil may resuspend into the air (Brahney et al., 2021; Jin et al., 2022). These pathways are all possible ways for MPs to be exposed to the human body, and controlling these

**Figure 5** Source-Pathway-Receptor model associated with three different MP and plasticizer sources.

## 3.5.2 Risk assessment of MPs

Based on the ecological risks of MPs for different sources (Table S6), Plastic Burning and Fruit-bag Burning were categorized as Level III (high risk). This may be attributed to the fact that PMMA, a compound with high hazard score, accounted for a higher proportion of MPs emitted from combustion sources. In contrast, Road Traffic, Agricultural Film, and Livestock Breeding sources, with lower hazard scores, were categorized as Level II (lower risk).

In this study, the health risks of MPs and plasticizers in PM<sub>2.5</sub> and PM<sub>10</sub> from five sources were analyzed as well. The total non-carcinogenic risk (HI) ranged from 1.36×10<sup>-4</sup> (Agricultural Film) to 5.20×10<sup>-4</sup> (Livestock Breeding) in PM<sub>2.5</sub> and 2.01×10<sup>-4</sup> (Road Traffic) to 8.96×10<sup>-4</sup> (Livestock Breeding) in PM<sub>10</sub>, inconsistent with the mass concentration ranking of MPs and plasticizers in various sources. All HI values of each source were significantly lower than the international safety threshold (HI=1). The highest HI was observed in Livestock Breeding, followed by Plastic Burning,

with values of 4.49×10<sup>-4</sup> and 8.73×10<sup>-4</sup> for PM<sub>2.5</sub> and PM<sub>10</sub>, respectively. Figure S2 illustrated the contributions of different compounds to HI. PAEs contributed most significantly to HI, accounting for more than 60% in most sources, especially in Livestock Breeding (93.6% and 92.9% for PM<sub>2.5</sub> and PM<sub>10</sub>, respectively). Among all compounds, one of PAEs, DEHP displayed the highest non-carcinogenic risk (Figure S2). In Road Traffic, BT and MBT exhibited the higher HI than other sources, with BT accounting for 20.3% (PM<sub>2.5</sub>) and 18.1% (PM<sub>10</sub>), followed by MBT (6.5% and 6.7%, respectively). Moreover, PS in Agricultural Film source exhibited the prominent HI values, with the proportion of 27.6% and 26.3% for PM<sub>2.5</sub> and PM<sub>10</sub>, respectively, 10-77 times higher than other sources. These findings emphasize the need to focus on PAE in Livestock Breeding, BT and MBT in Road Traffic and PS in Agricultural Film as a priority for MPs pollution control, aiming to minimize associated human non-carcinogenic risks.

ILCR for the three carcinogenic compounds (BT, BBP, and DEHP) were calculated in this study (Guyton et al., 2009; Ma et al., 2020; Liu et al., 2023). The ILCR values for each compound varied between 7.03×10<sup>-16</sup> and 1.77×10<sup>-7</sup>, which were all below the safety threshold (10<sup>-6</sup>). But this cannot be taken lightly, as there are many types of environmental pollutants, and their carcinogenic risks are additive and cumulative. Compared to other sources, Livestock Breeding had the highest total ILCR values (∑ILCR) (1.01×10<sup>-7</sup> in PM<sub>2.5</sub> and 1.8×10<sup>-7</sup> PM<sub>10</sub>), although the mass concentration of MPs and plasticizers in this source was not the highest. Combined with the HI results, we can see that Livestock Breeding emitted the higher concentrations of toxic MPs and plasticizers, increasing the human health risk. Comparison of the carcinogenic risks of different compounds showed that DEHP accounted for more than 97% of ∑ILCR in each source, which is

the species that needs to be controlled the most in this study.

3.5.3 Effect of MPs and plasticizers on oxidative potential

Figure S3 demonstrates the oxidative potential capacity of PM<sub>2.5</sub> and PM<sub>10</sub> from five sources. Overall, PM<sub>2.5</sub> exhibited a generally higher level of oxidative potential than PM<sub>10</sub>. The larger specific surface area of PM<sub>2.5</sub> can enhance its reactivity with DTT and facilitate ROS production (Boogaard et al., 2012; Feng et al., 2016; Chirizzi et al., 2017). Moreover, the presence of certain components in PM<sub>2.5-10</sub> may actually weaken the ability of PM<sub>2.5</sub> components to induce ROS production (Boogaard et al., 2012; Chirizzi et al., 2017). This may suggest a completely different mechanism for the generation of ROS between coarse and fine particles. Therefore, the results show that PM<sub>10</sub> has lower oxidative potential than PM<sub>2.5</sub> for mostly sources in this study (Plastic Burning, Fruit-bag Burning, Agricultural Film, and Livestock Breeding), which requires further research in the future.

 $PM_{2.5}$  from Fruit-bag Burning exhibited the highest oxidative potential with a value of  $77.0 \pm 59.8$  nmol min<sup>-1</sup> m<sup>-3</sup>, while its  $PM_{10}$  DTT value was only  $18.32 \pm 8.27$  nmol min<sup>-1</sup> m<sup>-3</sup>, indicating that oxidative potential of Fruit-bag Burning was mainly driven by  $PM_{2.5}$ . For Plastic Burning source, the DTT values of  $PM_{2.5}$  and  $PM_{10}$  were  $58.6 \pm 21.2$  and  $28.0 \pm 23.7$  nmol min<sup>-1</sup> m<sup>-3</sup>, respectively, both at relatively high levels. In contrast,  $PM_{2.5}$  from road sources showed a low oxidative potential  $(0.75 \pm 0.09 \text{ nmol min}^{-1} \text{ m}^{-3})$ , and Road Traffic was the only source with a higher oxidative potential for  $PM_{10}$  than  $PM_{2.5}$ . This is likely attributed to the unique characteristics of road dust, which is rich in coarse particles (Boogaard et al., 2012; Pant et al., 2015; Shirmohammadi et al., 2017). Road dust contains a high concentration of metal compounds, catalyzing the DTT

consumption (Shirmohammadi et al., 2017). Future research should focus on the size dependency of oxidative potential for different sources, as it has significant implications for health impact.

To investigate the impact of MPs and plasticizers on oxidative potential, spearman correlation analysis was employed to assess the relationships between these compounds and DTT. As shown in Figure 6, PMMA (R=0.77, P

## Figure 6 Correlations between DTT, MPs, and plasticizers (\*P < 0.05; \*\*P < 0.01)

## 4. Conclusion

In this study, the five typical plastic emission sources in the Guanzhong Plain, China were selected to investigate the characteristics of MPs and plasticizers in PM2.5 and PM2.5-10. The concentration levels of MPs and plasticizers in combustion sources (Plastic Burning and Fruit-bag Burning) were higher than non-combustion sources (Road Traffic, Agricultural Film, and Livestock Breeding), highlighting the necessity of tightening plastic combustion regulations to address atmospheric MPs pollution. Most detected MP and plasticizer were more abundant in PM<sub>2.5</sub> than PM<sub>2.5-10</sub> for most sources. Plastic Burning source is recognized by high loadings of HOBT, PMMA, and DEP. Fruit-bag Burning exhibited the high abundances of DnOP and higher in PM<sub>2.5-10</sub> than PM<sub>2.5</sub>. Since tire wear particle is one of the main sources of Road Traffic MPs, rubber compositions (NR, BR+SBR) accounted for the highest proportions. Agricultural Film is mainly characterized by high abundance of PS. The high proportions of OBS and CBS can distinguish Livestock Breeding from the other sources and there are still many unknown aspects of Livestock Breeding sources that require future research attention. This study develops a complete eco-health risk assessment system, identifying combustion sources (Plastic Burning and Fruit-bag Burning) as the high ecological risk emitters, Livestock Breeding as the high health risk contributor, and DEHP as a key health damage pollutant due to its combined non-carcinogenic risk, carcinogenic risk, and oxidative potential effects.

Our results could contribute to provide a scientific foundation for accurately identifying the sources and risks of atmospheric MPs, and developing efficient management strategies. The single-

shot pyrolysis protocol used in detecting MPs in this study enables quasi-instantaneous and homogenous pyrolysis (Seeley and Lynch, 2023). However, the complex products of combustion process may lead to interference from non-polymeric components during single pyrolysis. Therefore, future work should systematically compare different pyrolysis approaches in order to improve accuracy of MPs for complex samples. Additionally, future studies should expand the range of assessed MPs and plasticizers and integrating multiple ecological health assessment methods to further refine the health risk assessment system and deepen the understanding of the environmental and health hazards of MPs.

Data availability

- Data will be made available on request.
- Declaration of competing interest
- The authors declare that they have no known competing financial interests or personal relationships that could have appeared to influence the work reported in this paper.
- Supplementary data
- Supplementary data in this manuscript can be found in Supporting Information.
- Author contribution
- LL: Data Curation, Formal analysis, Writing-Original draft preparation; HX: Supervision,
- Project administration, Funding acquisition, Writing-Original draft preparation; MY: Data Curation,
- Investigation; TH: Supervision, Investigation; AA: Investigation, Data Curation; JS: Investigation;
- ZS: Review & Edit, Supervision.
- Acknowledgments

- This research was supported by Shaanxi Province Natural Science Basic Research Program
- (2025JC-QYCX-030) and the open fund of the State Key Laboratory of Loess Science, Institute of
- Earth Environment, Chinese Academy of Sciences (SKLLOG2425). We would like to express our
- sincere gratitude to Hongmeng Reference Material for their generous support.

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
