# Peer review of "Insight into the size-resolved markers and eco-health"

_EGUsphere, 2025_

## Author Comment (AC2)

**Reviewer 2:**

This paper investigates emitted microplastics aerosols from five sources in northwest China. The authors present a comprehensive characterization of the sources and plasticizer profiles. An eco-health risk assessment was conducted, providing key details like daily exposure risks. These findings are especially valuable for informing risk-mitigation policies and protecting individuals who are often close to sources like those in rural households which burn plastics, agricultural workers or incineration plant workers. This paper merits publication in ACP after some minor revisions.

Response:

We sincerely appreciate the time and effort you have dedicated to reviewing our manuscript. We also thank you for your meticulous attention to detail, which has helped us improve the clarity and accuracy of the manuscript. We have carefully addressed each suggestion and implemented the necessary revisions. Below are our detailed responses to your comments:

General comments

The authors must be aware of using past and present tense. Many places are inconsistent with using the correct tense.

Response:

Thank you for your suggestion. We have conducted a systematic review of verb tenses throughout the manuscript. Specifically, we have used the past tense throughout the sections on experimental methods and results to accurately reflect completed actions. In the discussion section, we have employed the present tense to describe current interpretations and implications of the findings.

1. Line 19: Add "i.e.," in front of phthalates

Response:

Suggestion taken.

2. Add a comma after (AF).

Response:

Suggestion taken.

3. Replace the verb "features".

Response:

This point has been revised as follows.

*"PB exhibits high proportions of poly(methyl methacrylate) (PMMA) and 2-hydroxy benzothiazole (HOBT), with PMMA being more abundant in coarse particles ($PM_{coarse}$)."*

4. Line 36: Spell out ROS to keep consistency.

Response:

Suggestion taken.

5. Line 43: in a range of "five millimeters to one micrometer"

Response:

Suggestion taken.

6. Line 53: It is recommended to change "results" to "result" because the subject is plural.

Response:

Suggestion taken.

7. Line 59: Define PMs.

Response:

The definition of PMs has been added.

*"Evangeliou et al. (2020) have estimated that annual total global tire wear particle emissions were 2907 kt $y^{-1}$, with 29 and 288 kt $y^{-1}$ for $PM_{2.5}$ (particulate matter with aerodynamic diameters $\leq 2.5$ μm) and $PM_{10}$ ($\leq 10$ μm), respectively."*

Reference:

Evangeliou, N., Grythe, H., Klimont, Z., Heyes, C., Eckhardt, S., Lopez-Aparicio, S., and Stohl, A.: Atmospheric transport is a major pathway of microplastics to remote regions, Nat. Commun., 11, 10.1038/s41467-020-17201-9, 2020.

8. Line 77: rephrase the words in the presentation.

Response:

The statement has been rewritten as below.

*"Airborne MPs can easily enter the human body directly via respiratory system respiration compared to other environmental exposure pathways, posing a direct and serious health concern (Liao et al., 2021; Luo and Guo, 2025)."*

Reference:

Liao, Z., Ji, X., Ma, Y., Lv, B., Huang, W., Zhu, X., Fang, M., Wang, Q., Wang, X., Dahlgren, R., and Shang, X.: Airborne microplastics in indoor and outdoor environments of a coastal city in Eastern China, J. Hazard. Mater., 417, 10.1016/j.jhazmat.2021.126007, 2021.

Luo, R. C. and Guo, K.: The hidden threat of microplastics in the bloodstream, The Innovation Life, 3, 10.59717/j.xinn-life.2025.100130, 2025.

9. Line 81-83: The sentence must be advanced.

Response:

We have advanced the sentence. The revised paragraph is as follows.

*"Recent studies suggest that these inhaled pollutants can promote reactive oxygen species (ROS) generation (Wang et al., 2024). Oxidative potential is a metric reflecting the ability of inhaled pollutants to produce ROS, serving as a critical indicator of PM toxicity (Jiang et al., 2019; Bates et al., 2019; Luo et al., 2024). The ROS overproduction acts as a central driver of oxidative stress, which can damage biomolecules and disrupt cellular functions (Bates et al., 2019; Jiang et al., 2019). Previous studies have demonstrated that metals and organic compounds can affect the oxidative potential of PMs (Luo et al., 2023; Ghanem et al., 2021)."*

Reference:

Wang, L. J., Pei, W. L., Li, J. C., Feng, Y. M., Gao, X. S., Jiang, P., Wu, Q., and Li, L.:

Microplastics induced apoptosis in macrophages by promoting ROS generation and altering metabolic profiles, Ecotox. Environ. Safe., 271, 11, 10.1016/j.ecoenv.2024.115970, 2024.

Jiang, H. H., Ahmed, C. M. S., Canchola, A., Chen, J. Y., and Lin, Y. H.: Use of dithiothreitol assay to evaluate the oxidative potential of atmospheric aerosols, Atmosphere, 10, 21, 10.3390/atmos10100571, 2019.

Bates, J. T., Fang, T., Verma, V., Zeng, L. H., Weber, R. J., Tolbert, P. E., Abrams, J. Y., Sarnat, S. E., Klein, M., Mulholland, J. A., and Russell, A. G.: Review of acellular assays of ambient particulate matter oxidative potential: Methods and relationships with composition, sources, and health effects, Environ. Sci. Technol., 53, 4003-4019, 10.1021/acs.est.8b03430, 2019.

Luo, L., Guo, S., Shen, D., Shentu, J., Lu, L., Qi, S., Zhu, M., and Long, Y.: Characteristics and release potential of microplastics in municipal solid waste incineration bottom ash, Chemosphere, 364, 143163, 10.1016/j.chemosphere.2024.143163, 2024.

Luo, Y., Zeng, Y. L., Xu, H. M., Li, D., Zhang, T., Lei, Y. L., Huang, S. S., and Shen, Z. X.: Connecting oxidative potential with organic carbon molecule composition and source-specific apportionment in PM2.5 in Xi'an, China, Atmospheric Environment, 306, 9, 10.1016/j.atmosenv.2023.119808, 2023.

Ghanem, M., Perdrix, E., Alleman, L. Y., Rousset, D., and Coddeville, P.: Phosphate Buffer Solubility and Oxidative Potential of Single Metals or Multielement Particles of Welding Fumes, Atmosphere, 12, 23, 10.3390/atmos12010030, 2021.

10. Line 84: Add "Most of the" at the start of the sentences.
Response:
Suggestion taken.

11. Line 91: Remove "primary"
Response:
Suggestion taken.

12. The First and Second paragraphs must be combined with a reasonable logic. It should start with the sample collection place and time.
Response:
Thank you for your valuable suggestion. We have restructured these paragraphs to improve logical flow by beginning with the spatiotemporal context of sample collection (location and time), then introducing the sampling methodology, and finally presenting the analytical approach. The revised paragraph starts as follows.

*"During January and February 2024, PM$_{2.5}$ and PM$_{10}$ samples were collected simultaneously from five distinct sources in three key cities of the Guanzhong Plain: Xi'an, Tongchuan, and Xianyang (Figure S1). The selected sources included Plastic Burning (PB), Fruit-bag Burning (FB), Road Traffic (RT), Agricultural Film (AF), and Livestock Breeding (LB)."*

13. Line 102: The term of "PB burned plastics" is not an appropriate term.
Response:
It has been revised as follows.

*"It should be noted that the types of plastics used for Plastic Burning source incineration*

*including plastic bags, bottles, disposable tableware, foam boxes, and other plastic daily necessities."*

14. Table 1: The format of the table must be advanced. The alignment of the first line in each block must be on the top.

Response:

Suggestion taken. The revised Table 1 is as follows:

Table 1 Basic sampling information of target emission sources

| Emission source | Sampling duration (h) | Sampling height | Sample No. | Sampling location |
|---|---|---|---|---|
| Plastic burning (PB) | 2.0 | 3-4 m above the ground | 5 | Open space, about 1 m downwind of chimney of rural household stove in rural |
| Fruit bag burning (FB) | 2.0 | 3-4 m above the ground | 5 | Xianyang |
| Road Traffic (RT) | 13.6-14.1 | 3 m above the ground | 5 | Open space, flyovers on traffic arteries in downtown Xi'an |
| Agricultural film (AF) | 24 | 1.5 m above the ground | 5 | Open space, about 2 m away from the greenhouse in farmland in rural Tongchuan |
| Livestock breeding (LB) | 2.5-3.5 | 1.5 m above the ground | 5 | About 1 m from the feed trough in a cow shed of approximately 8 $m^2$ in rural Tongchuan |

15. Line 122" A missing space for 'at -20°C'.

Response:

Suggestion taken.

16. Figure 1: It is recommended that the error bars should be thickened to increase the clarity.

Response:

Suggestion taken. The modified Figure 1 is as follows:

[Figure]

**Figure 1** Average concentrations of MPs (a) and plasticizers (b) in PM$_{2.5}$ and PM$_{2.5-10}$ from five

sources (PB: Plastic Burning, FB: Fruit-bag Burning, RT: Road Traffic, AF: Agricultural Film, LB: Livestock Breeding).

Section 2.2.

17. Line 147: More advanced information on the procedures of the GC/MS analysis must be shown (i.e. capillary column used, and MSD setting), not just referring to a reference.

Response:

We sincerely appreciate your suggestion. As requested, we have now provided the detailed GC/MS analytical procedures, including capillary column, MSD settings, and other relevant parameters, in Supplementary Information (Appendix 2). The specific information is as follows:

*"**Appendix 2** Analysis of plasticizers*

*Phthalates were quantified using in-injection port-thermal desorption/mass spectrometry (TD-GC/MS) method. Aliquots of the filters (1.578 cm$^2$) were cut into small pieces, spiked with ISs (Chrysene-d12, 96%, LGC Standard Limited, United States), and inserted into thermal tubes (78 mm long, 4 mm I.D., 6.35 mm O.D., Agilent Technology, USA) for analyses. The sample tube was directly loaded into a GC injection port (GC7890, Agilent Technology), at an initial temperature of 50°C. The temperature of injector was then ramped to 275°C for desorption in a splitless mode, while the GC oven temperature was kept at 30°C. The desorbed analytes were refocused at the column head. After the injector temperature reaches the set point, the oven program starts. The analytes were speared by an DB-5ms capillary column (30 m × 0.25 mm i.d. × 0.25 μm film thickness; J&W Scientific). The carrier gas was ultra-high purity (99.9999%) He at a constant flow of 1.0 cm$^3$ min-1. The MSD (5975, Agilent Technology) was full scanned from 50 to 550 amu under electron impact ionization (EI) at a voltage of 70 eV and an ion source temperature of 230°C. Identification was achieved by characteristic ion and retention times of the chromatographic peaks with those of authentic standards.*

*To quantify benzothiazole and its derivatives, each of 1.578 cm$^2$ of the filter sample was cut and spiked with an internal standard (IS) of benzothiazole-d4 (benzothiazole=d4, 95%, LGC Standard Limited, United States). The filter is transferred into a test tube, and 10 mL of a mixture of ultrapure deionized water (18 M-Ohm) and methanol (HPLC grade, Fisher Chemical, USA) (5:3, v:v) was added. The sample was extracted in an ultrasonic water bath at room temperature for 60 min. The combined extracts were concentrated and then diluted to with ultrapure deionized water containing 0.2% v/v formic acid (~pH 2.5). The diluted extract was purified using an Oasis HLB Flangeless Vac Cartridge (3cc, 60 mg sorbent per cartridge, 30 μm particle size; Waters, USA). The target analytes were eluted with 5 mL of methanol and the eluents were evaporated to 1 mL under a gentle nitrogen stream prior to analysis. The separation of target analytes was performed using an ultra-performance liquid chromatography (UPLC; ACQUITY, Waters), and both identification and quantification were accomplished using a triple quadrupole mass spectrometer (ESI-MS/MS; Xevo TQ-S, Waters). An ACQUITY UPLC BEH SHIELD RP 18 column (100 mm ×3 mm × 1.7 mm) was serially connected to a Vanguard column (BEH C18, 5 mm × 2.1 mm × 1.7 mm). The mobile phase comprises 100% methanol (A) and ultrapure deionized water acidified with 0.1% (v/v) formic acid (B) at a flow rate of 450 mL min$^{-1}$. A gradient elution program was applied for the separation. The tandem MS system was operated in the positive ion multiple reaction monitoring mode. Identification was achieved by characteristic ion and retention times of the chromatographic peaks with those of authentic standards."*

18.  More detailed QC/QA procedures must be given for all analyses.

Response:

Thank you for your suggestion. The details of QC/QA procedures have already been documented in Section 2.3 of the Methods and as follows.

*"Quality assurance/Quality control (QA/QC)*

*The flow rates of all samplers were calibrated using a mass flowmeter (Model 4140, TSI, Shoreview, MN, USA) before and after each sampling cycle. All quartz filters used in this study were preheated at 800°C for 3 h to remove any potential contaminants and then cooled before use. To minimize experimental error, sampling was conducted in duplicate for each particle size of each source. For the chemical measurements, one in every 10 samples was reanalyzed for quantity assurance purposes, and the SD errors of replicate trials were within 10% for the pyrolysis analyses. Calibration curves were established using reference standards. The linearities of the standard calibration curves were > 0.987. The standard deviations of the pyrolyzed standard were within 94.1% to 98.3%. Background contamination (Table S3) was monitored by processing operational blanks (unexposed filters) simultaneously with field samples."*

19.  Section 2.5

Consider renaming it as "Data Analysis and Statistical Method.

Response:

Suggestion taken.

20. Line 216: The data must be tabulated instead of shown in a figure. The values are more appropriate to demonstrate the actual circumstances.

Response:

Thank you for your valuable suggestion. We have added Table S2 in the Supporting Information (SI) file.

21.  The values and percentages must be presented in the form of mean plus standard deviation. Check out all these in all sections.

Response:

Thank you for your valuable comments. We have revised all values to the format of mean $\pm$ standard deviation throughout the text.

22.  The uses of PM10 and PMcoarse are confusing Standardize to use one term.

Response:

Sorry for the confusion. We have standardized our terminology by replacing "$PM_{coarse}$" with "$PM_{2.5-10}$" (particulate matter with aerodynamic diameters between 2.5 and 10 μm). When characterizing microplastics and plasticizers, we specifically use "fine particles ($PM_{2.5}$)" and "coarse particles ($PM_{2.5-10}$)" to clearly differentiate and compare their characteristics across different size fractions.

However, we need to retain "$PM_{10}$" in the sections of the source profiles and health risk assessments of MPs and plasticizers. For the source profile and health risk assessment sections, we primarily use "$PM_{10}$" because it represents the total inhalable particulate matter that is most

relevant for exposure assessment and it allows direct comparison with other studies.

23. Line 230: The verb "crushed" is inappropriate.

Response:

This point has been revised as follows.

*"One possible explanation for this is that plastic waste can be fragmented into MPs during the process of combustion."*

24. Line 274: What is the meaning of "greenhouses" here?

Response:

Thank you for your question. In original Line 274, the term "greenhouses" refers to controlled agricultural environments, plastic film structures covering crops, which are used to regulate climatic conditions. In this study, AF (agricultural film) primarily refers to greenhouse coverings, which are generally made of transparent polyethylene and are typically less than 8 μm thick (Wang et al., 2018; Li et al., 2022). The compositions of MPs in AF may be influenced by wear/degradation of plastic film and other plastic equipment in greenhouses. The original lines 272-274 suggest that the high PS content in AF could stem from PS-based materials (e.g., lamp-chimneys, electrical devices) commonly used in greenhouse facilities.

To avoid ambiguity, we revised the original sentence (Lines 272-274) as follows:

*"Agricultural facilities made of PS (e.g., lamp-chimneys, electrical devices) in greenhouses may influence the MPs composition of Agricultural Film source (Qi et al., 2023)."*

Reference:

Qi, R., Tang, Y., Jones, D. L., He, W., and Yan, C.: Occurrence and characteristics of microplastics in soils from greenhouse and open-field cultivation using plastic mulch film, Sci. Total Environ., 905, 10.1016/j.scitotenv.2023.166935, 2023.

Wang, Y., Zhu, H. K., and Kannan, K.: A Review of Biomonitoring of Phthalate Exposures, Toxics, 7, 28, 10.3390/toxics7020021, 2019.

Wang, L., Zhang, B., & Tian, G. Q. (2018). Research on government intervention in agricultural plastic film using and recycling in Chinese. Issues in Agricultural Economy, 08, 137–144.

25. Line 281-284: The presentation could not be fully understood.

Response:

We apologize for the confusion in the presentation. The text has been revised for clarity as follows:

*"The percentages of BTs and BPA among the three detected plasticizer types were below 2%. The highest concentrations of PAEs, BTs, and BPA still appear in PB among five sources."*

26. Line 286: Relative: is an inappropriate word used here.

Response:

It has been revised as follows.

*"Compared to other sources, Road Traffic source demonstrated a higher concentration of BTs."*

27. Line 297: Is it should be shown in a new Sub-Section? Please verify that.

Response:
We do agree with the reviewer's comment. We have moved the content ("Compositions and distributions of PAEs", "Compositions and distributions of BTs") to a new Sub-Section.

28. Figure 5 should use legends in the figure to clarify all symbols rather than explanation in the captions ("sections marked in red")

Response:
Thank you for your comment. We have revised Figure 5 to incorporate legends within the figure. The revised Figure 5 is as follows.

[Figure]

**Figure 5** Source-Pathway-Receptor model associated with three different MP and plasticizer sources.

29. The DTT included in this study was reported in volume-base. What is the reason for choosing this DTT to describe the toxicity of PMs?

Response:
We adopted the volume-based DTT assay in this study to maintain consistency with the volume-based quantification of MPs and plasticizers in $PM_{2.5}$ and $PM_{10}$ (quantified as mass per cubic meter of air). This approach facilitates a clearer interpretation of the relationship between the concentrations of MPs/plasticizers and the oxidative potential of PMs per unit volume of air ($m^3$).

30. The future research directions must be advanced. For example, it is recommended to consider coupling the multiple ecological health assessment methods mentioned in this study, measure the weights of different methods, and provide comprehensive evaluation indices.

Response:
Thank you for your valuable suggestion. We have revised the relevant sentence in the manuscript to incorporate the integration of multiple ecological health assessment methods.
*"Future studies should expand the range of assessed MPs and plasticizers and integrating multiple ecological health assessment methods to further refine the health risk assessment system and deepen the understanding of the environmental and health hazards of MPs."*

---

## Author Comment (AC3)

**Reviewer 3:**

Liu et al. investigated the characteristics, source profiles, and health risks of airborne microplastics and plasticizers emitted from plastic burning, fruit bag burning, road traffic, agricultural film, and livestock breeding sources in Guanzhong Plain, Northern China. Knowledge on the characteristics of atmospheric microplastics in Northern China is currently limited. This study provides important insights for source tracking and risk assessment of these emerging contaminants. I recommend this paper to be published after minor revisions.

Dear reviewer,

Thanks very much for taking your time to review this manuscript. We sincerely appreciate your constructive suggestions for improvement. We have carefully considered each of your comments and have revised the manuscript accordingly to improve the quality of our manuscript.

1. Many plasticizers were used in the manufacturing of plastics. The focus of this study is PAEs, BPs and BPA. Please provide the reason for selecting these three groups of plastics to study.

Response:

The selection of BTs, PAEs, and BPA as the focus of our study is driven by their widespread use, ubiquitous in the environment and potential health risks.

Phthalate esters (PAEs) are the most widely used plasticizers globally, dominating the plastic additive market. He et al. (2020) demonstrated that during 2007-2017, the annual global production of PAEs increased from 2.7 million tons to 6 million tons. Moreover, China is recognized as the largest importer of PAEs worldwide. Benzothiazoles (BTs) are extensively used in automotive tires and agrochemicals. High concentrations of BTs were discovered in the street runoff, suggesting that these tire material-related compounds can persevere in the environment (Zhang et al., 2018). Exposure to BTs may result in central nervous system depression, liver and kidney damage, dermatitis, and pulmonary irritation (Ginsberg et al., 2011). Bisphenol A (BPA) as a common industrial chemical component in many products, has steadily grown over the last 50 years (Corrales et al., 2015). Growth of global production has consistently ranged between 0% and 5% annually (Corrales et al., 2015). PAEs and BPA considered as endocrine disruptors, are demonstrated to impair reproductive function and development in laboratory animals (Wang et al., 2018).

Despite the well-documented health risks associated with these plasticizers in laboratory settings, there is a significant gap in understanding their real-world emissions and health impacts. In this study, we aim to fill this gap by investigating the emission characteristics of these plasticizers from various sources and evaluating their potential health impacts based on real-world concentration levels.

The related sentences have been revised as follows.

[revised manuscript text omitted]

2. The authors mention ROS formation in the text for several times. You use DTT consumption rate to represent oxidative potential. Note that DTT consumption rate is not equal to ROS formation. Some chemicals can consume DTT without the formation of ROS during interaction.

Response:

Thank you for the reviewer's insightful comments. DTT consumption rate indeed reflects oxidative potential rather than directly measuring ROS formation. We have carefully revised the manuscript to consistently use the term "oxidative potential (OP)" rather than "ROS formation." The revised paragraph is as follows:

*"Figure S3 demonstrates the oxidative potential capacity of $PM_{2.5}$ and $PM_{10}$ from five sources. Overall, $PM_{2.5}$ exhibits a generally higher level of oxidative potential than $PM_{10}$."*

3. It is interesting that PMMA, PET, PE, PAEs, and BPA showed a positive correlation with oxidative potential. I encourage the authors to further discuss the possible mechanism which would help readers to understand their health impact.

Response:

The observed positive correlation between polyethylene terephthalate (PET) and oxidative potential (OP) may be mechanistically explained through environmentally persistent free radicals (EPFRs) (Zhang et al., 2024). EPFRs can form in MPs with conjugated aromatic-ring structures under thermal or photochemical aging conditions (e.g., high temperature/solar irradiation) (Yuan et al., 2022; Zhu et al., 2019). Previous studies have demonstrated that OP of particulate matter exhibits positive correlations with the concentration of EPFRs (Huang et al., 2021; Li et al., 2023). EPFRs can directly participate in redox reactions or mediate reactive oxygen species (ROS) generation to enhance the consumption of DTT (Yang et al., 2024). Therefore, PET and other MPs showing a positive correlation with OP may be attributed the formation of EPFRs.

Previous studies (Yu et al., 2022; Lin and Yu, 2019) have shown that organic compounds with a similar structure to PAEs (containing a benzene ring) can form complexation with metal ions (e.g., $Mn^{2+}$, $Fe^{2+}$, etc.), and this complexation can accelerate the consumption of DTT. Since atmospheric particles contain a certain amount of metal ions, we speculate that the strong

correlation between PAEs and the OP of PM may be due to this reason.


Response:

We did consider the effect of particulate matter particle size in our risk assessment. Specifically, we compared the difference in risk indices for PM10 and PM2.5. As shown in Figure S2, for the same source, the difference in non-carcinogenic risk for different particle sizes is small. This suggests that the effect of particle size on MPs induced health risk is relatively limited. Therefore, in the Discussion section, we focused more on comparing the differences in health risk between sources rather than particle sizes.

[Figure]

**Figure S2** The non-cancer risks of MPs and plasticizers from five sources (PB: Plastic Burning, FB: Fruit-bag Burning, RT: Road Traffic, AF: Agricultural Film, LB: Livestock Breeding).

5. Line 51, Agricultural activities are also a significant contributor to atmospheric MPs?

Response:

Thank you very much for your valuable comment. Although agricultural activities are not a direct source of microplastics (MPs), they serve as a significant indirect contributor to atmospheric MPs. due to the use of plastic mulch films and other related plastic products, as well as the deposition of MPs in agricultural soils transported via atmospheric and surface runoff pathways, MPs are present in the soil (Lakhiar et al., 2024). Agricultural activities, by increasing soil disturbance, may cause the resuspension of MPs into the atmosphere. In this way, agricultural activities indirectly contribute to atmospheric MPs. For example, the paper "Activities of Microplastics (MPs) in Agricultural Soil: A Review of MPs Pollution from the Perspective of Agricultural Ecosystems" clearly indicates that agricultural machinery operations, such as plowing and harvesting, can disturb the soil and expose MPs to the air. These particles can then be released into the air by wind.

---

## Author Comment (AC4)

**Reviewer 4:**

In this study, particles emitted from five typical sources (plastic burning, fruit bag burning, road traffic, agricultural film, livestock breeding) in the Guanzhong Plain, northwest China were collected. The authors investigate characteristics and eco-health risks of microplastics (MPs) and plasticizers (phthalates, benzothiazoles, bisphenol A) in PM2.5 and PM10. It identifies source-specific markers and reveals that combustion-derived MPs pose the highest ecological risks, while livestock breeding and plastic burning exhibit elevated human health hazards. The topic is within the scope of ACP. Overall, I recommend this paper to be accepted after revisions.

Response:

We sincerely appreciate the time and effort you have dedicated to reviewing our manuscript. We have carefully considered each of your suggestions and have made the necessary revisions to improve the quality of our manuscript.

1. Please be aware of the tense.

Response:

Thank you for your suggestion. We have conducted a systematic review of verb tenses throughout the manuscript. Specifically, we have used the past tense throughout the sections on experimental methods and results to accurately reflect completed actions. In the discussion section, we have employed the present tense to describe current interpretations and implications of the findings.

2. Please define PMcoarse at its first appearance in Abstract and Introduction. Also, maybe PM2.5-10 could be better for understanding than PMcoarse.

Response:

Thank you for the suggestion. We have standardized our terminology by replacing "$PM_{coarse}$" with "$PM_{2.5-10}$" and defined $PM_{2.5-10}$ at its first appearance in Abstract and Introduction.

*"PMMA being more abundant in $PM_{2.5-10}$ (aerodynamic diameters between 2.5 and 10 μm)."*

3. In this paper, five sources of MPs are analyzed, but excluding emerging sources like industrial emissions, construction, or textile abrasion, which may contribute significantly to atmospheric MPs.

Response:

Our study specifically focuses on outdoor environmental sources directly linked to daily human activities and exposure (living sources), particularly in understudied rural areas in Northwestern China. To maintain this focus, all sampling sites, except for road traffic source, were located in rural areas. The sampling for road traffic source was done in city to capture MPs from high traffic emission intensity, replacing the weak emission state in rural areas. But it cannot be denied that there are also certain vehicle emissions in rural areas.

Sources like industrial emissions, construction, while important, were considered less representative of the immediate, daily-life exposure pathways central to our research. That is to say, these sources that are far from our daily lives may contribute less to the health effects caused by MPs. As for textile abrasion, it is indeed a significant source of indoor MPs. We will definitely include this in our future research. Thank you for the valuable suggestion.

4. It is recommended that a separate QC/QA section can be added in Methods section, which

could provide more QC/QA details for collection and analyses.

Response:

Thank you for your suggestion. The details of QC/QA procedures have already been documented in Section 2.3 of the Methods.

*"Quality assurance/Quality control (QA/QC)*

*The flow rates of all samplers were calibrated using a mass flowmeter (Model 4140, TSI, Shoreview, MN, USA) before and after each sampling cycle. All quartz filters used in this study were preheated at 800°C for 3 h to remove any potential contaminants and then cooled before use. To minimize experimental error, sampling was conducted in duplicate for each particle size of each source. For the chemical measurements, one in every 10 samples was reanalyzed for quantity assurance purposes, and the SD errors of replicate trials were within 10% for the pyrolysis analyses. Calibration curves were established using reference standards. The linearities of the standard calibration curves were > 0.987. The standard deviations of the pyrolyzed standard were within 94.1% to 98.3%. Background contamination (Table S3) was monitored by processing operational blanks (unexposed filters) simultaneously with field samples."*

5. Section 2.2, what is the size of the filter membranes you cut for pyro-GC/MC analysis?

Response:

Thank you for your question. We cut filters with areas of 0.526 cm²(one circular punch from the filter) for MPs and 1.578 cm²(three circular punches from the filter) for each of the three plasticizers, respectively. We have added this detailed preparation information in Appendix 2 in Supplementary Information (SI) file.

*"In the quantification of microplastic (MPs) contents, 0.526 cm² of a filter sample was folded with ferromagnetic pyrofoil (F670, Japan Analytical Industry Co., Ltd, Tokyo, Japan) using a clean flip head tweezer."*

*"To quantify each of the three plasticizers, 1.578 cm² of the filter sample was cut into small pieces."*

6. Figure 2, the caption of Y-axis, is it the proportion of MPs in "PMs"? Or "MPs".

Response:

Thank you for pointing this out. Sorry for the mistake. The caption of Y-axis is the proportion of MPs in total MPs (%). Figure 2 has been modified as below.

[Figure]

**Figure 2** Chemical composition of microplastics in $PM_{2.5-10}$ and $PM_{2.5}$ from the five sources (PB: Plastic Burning, FB: Fruit-bag Burning, RT: Road Traffic, AF: Agricultural Film, LB: Livestock Breeding).

7. Figure 2, what does the slash lines stands for?

Response:

The slash lines are used to clearly distinguish the differences in the distribution of microplastics between $PM_{2.5}$ and $PM_{2.5-10}$.

8. Line 198, the health risk assessment uses U.S. EPA-derived exposure parameters, which may not fully align with Chinese population characteristics.

Response:

Thank you for your comments. In our study, the reference dose (RfD) and slope factor (SF) were derived from U.S. EPA data, as these values are widely recognized and standardized, providing a consistent and representative basis for health risk assessments globally. These parameters are also fundamental properties of chemical species and are almost unaffected by nation and regions.

However, parameters such as exposure time and exposure frequency were estimated based on the activity patterns of the population in the Guanzhong region. This point allowed us to incorporate local characteristics into our research. In future research, we will continue to refine our methods by integrating more localized exposure parameters to better reflect the behaviors of the Chinese population.

9. References are sometimes listed without page ranges. Also, there are some wired words (i.e., PM2.5) in Line 614 and 617.

Response:

Sorry for the errors. The same errors have been revised in the manuscript.

10. Introduction section Line 45, please consider adding these references: Micro/nanoplastics in the Shenyang city atmosphere: Distribution and sources, Environ. Pollut., 2025, 372, 126027; Characterization of atmospheric microplastics in Hangzhou, a megacity of the Yangtze river delta, China, Environ. Sci.: Atmos., 2024, 4, 1161-1169.

Response:

The authors have added the references in Lines 63 and 67.

---

## Author Comment (AC5)

**Reviewer 5:**

This paper describes the microplastics in the coarse and fine aerosol fractions from five different sources in northern China, quantified using pyrolysis GC/MS. A selection of common plastic additives was also detected. The oxidative potentials of filter extracts were determined using a DTT assay. Livestock breeding and plastic burning were found to produce the largest potential hazards of the five sources tested. This work represents an important study in a growing area of interest in atmospheric science, and I would recommend publication if the authors can add clarity to their methodology, QA/QC protocols, and data analysis.

Dear reviewer,

Thanks very much for taking your time to review this manuscript. We have carefully reviewed your comments and made the appropriate revisions to improve the quality of our manuscript. We have already added more details to clarify our methodology, QA/QC protocols, and data analysis. Below are our detailed responses to your comments:

Specific comments:

Line 39 "Global plastic production has gradually increased" – has it gradually or exponentially increased?

Response:

Thank you for highlighting this important distinction. Based on data from "Production, use, and fate of all plastics ever made (Sci. Adv., 2017, 3, e1700782)", we observed that global plastic production growth accelerated significantly after 1990, with a nearly vertical rise post-2000. Thus, "exponentially increased" more accurately reflects the trend of global plastic production than "gradually increased". We have revised the sentence as follows.

*"Global plastic production has increased exponentially after 1990, resulting in serious environmental contamination."*

Reference:

Geyer, R., Jambeck, J. R., and Law, K. L.: Production, use, and fate of all plastics ever made, Sci. Adv., 3, 5, 10.1126/sciadv.1700782, 2017.

Line 43 "Current research on MPs pollution sources have primarily focused on aquatic and terrestrial ecosystems.." – I would argue that current research has started to focus much more on airborne plastics, thus, I would recommend rewording this line.

Response:

We do agree with the reviewer's comment. This point has been revised as follows:

*"Research on microplastics (MPs) pollution initially focused on aquatic and terrestrial ecosystems, but recent years have seen growing attention to atmospheric MP pollution (Allen et al., 2020)."*

Reference:

Allen, S., Allen, D., Moss, K., Le Roux, G., Phoenix, V. R., and Sonke, J. E.: Examination of the ocean as a source for atmospheric microplastics, Plos One, 15, 10.1371/journal.pone.0232746, 2020.

Line 50 – I would add a qualifier in this line as the Yang et al. (2021) study used a cutoff size of

50 μm, thus, there are likely many more plastic particles produced than the values reported in the study.

Response:

Thank you for pointing this out. To clarify this limitation, we have added a qualifier to the sentence. The statement has been revised as follows.

*"Yang et al. (2021) have estimated that per metric ton of plastic can **potentially** produce 360 to 102,000 MPs, primarily composed of polypropylene (PP) and polystyrene (PS)."*

Reference:

Yang, Z., Lu, F., Zhang, H., Wang, W., Shao, L., Ye, J., and He, P.: Is incineration the terminator of plastics and microplastics? J. Hazard. Mater., 401, 10.1016/j.jhazmat.2020.123429, 2021.

Line 99 – Section 2.1, can the authors provide sampling dates and a map of where sampling took place (with long and lat)? This would be very helpful for future modelling studies.

Response:

We have revised the Figure S1 to indicate the sampling locations, sampling dates, and respective longitudes and latitudes for different sources (PB: Plastic Burning, FB: Fruit-bag Burning, RT: Road Traffic, AF: Agricultural Film, LB: Livestock Breeding).

[Figure]

**Figure S1** Sampling sites for different sources (PB: Plastic Burning, FB: Fruit-bag Burning, RT: Road Traffic, AF: Agricultural Film, LB: Livestock Breeding).

Line 121 – what was the material of the air cassettes used? And did it influence the final results?

Response:

The air cassettes used in our study were made of glass, a material selected for its chemical inertness to ensure no interference with experimental results. Each cassette was thoroughly rinsed with ethanol absolute before use.

Line 122 – did filter weighing follow any published standard and was static electricity neutralised prior to weighing?

Response:

The filter weighing process in our study was conducted in accordance with the Chinese National Standard GB/T 39193-2020 and the reference "Chemical characterization of $PM_{2.5}$ in heavy polluted industrial zones in the Guanzhong Plain, northwest China: Determination of fingerprint source profiles (Sci. Total Environ., 2022, 840, 156729).", which provide detailed protocols for the determination of particulate matter mass concentration in ambient air, ensuring the reliability of our filter weighing procedures. We used anti-static instrument (ANTIST-kit-un, MettlerToledo, Switzerland) to neutralize static electricity prior to weighing the filters.

Reference:

Wang, Z. X., Xu, H. M., Gu, Y. X., Feng, R., Zhang, N. N., Wang, Q. Y., Liu, S. X., Zhang, Q., Liu, P. P., Qu, L. L., Ho, S. S. H., Shen, Z. X., and Cao, J. J.: Chemical characterization of $PM_{2.5}$ in heavy polluted industrial zones in the Guanzhong Plain, northwest China: Determination of fingerprint source profiles, Sci. Total Environ., 840, 9, 10.1016/j.scitotenv.2022.156729, 2022b.

Line 124 – please provide more detailed information on how the field blanks were treated/collected.

Response:

We have provided the information on how the field blanks collected in Lines 163-166.

*"The field blank of each type of source was synchronously collected with active sampling. Unused filters (the same batch as sampling filters) were loaded into identical sampling devices, which were placed adjacent to operational samplers for the entire duration of one sampling event."*

Line 125 – Several studies have found that nitrile gloves can potentially influence pyrolysis GC/MS analysis of polymers, particularly PE. Please discuss.

Response:

We sincerely appreciate the reviewer's insightful comment. We fully acknowledge this concern and have taken multiple precautions in our experiment to minimize any possible interference.

Firstly, we cleaned the gloves with ethanol absolute before the experiment. This effectively removed potential contaminants from the gloves, minimizing their interference with the pyrolysis GC/MS analysis.

Secondly, we used stainless steel tweezers to handle the filter membrane throughout the experiment, avoiding direct contact between the gloves and the filters. This further ensured the reliability of the analysis results.

Moreover, the blank experiment was also conducted with gloves on, and the results from the blank experiment have been subtracted. This step effectively eliminated the potential interference that

might be caused by the gloves, ensuring the accuracy of the final results.

Line 138 – Please provide explicit details of preparation of samples, instrument configurations, and QA/QC protocols in this manuscript rather than referring to another paper. In regards to QA/QC, please provide information on positive and negative controls, the amount of analytes detected in background samples, and any subtractions that were completed to produce the final values. Addition to the supplemental would be a satisfactory placement for this information.

Response:

We do agree with the reviewer's comment. We have now incorporated the QA/QC protocols into Section 2.3 of the Methods. Additionally, the detailed sample preparation procedures and instrument configurations are provided in the Supplemental Information. The specific information is as follows:

*"Quality assurance/Quality control (QA/QC)*

*The flow rates of all samplers were calibrated using a mass flowmeter (Model 4140, TSI, Shoreview, MN, USA) before and after each sampling cycle. All quartz filters used in this study were preheated at 800°C for 3 h to remove any potential contaminants and then cooled before use. To minimize experimental error, sampling was conducted in duplicate for each particle size of each source. For the chemical measurements, one in every 10 samples was reanalyzed for quantity assurance purposes, and the SD errors of replicate trials were within 10% for the pyrolysis analyses. Calibration curves were established using reference standards. The linearities of the standard calibration curves were > 0.987. The standard deviations of the pyrolyzed standard were within 94.1% to 98.3%. Background contamination (Table S3) was monitored by processing operational blanks (unexposed filters) simultaneously with field samples."*

*"**Appendix 1** Analysis of Microplastics*

*In the quantification of miccroplastic (MPs) contents, 0.526 cm$^2$ of a filter sample was folded with ferromagnetic pyrofoil (F670, Japan Analytical Industry Co., Ltd, Tokyo, Japan) using a clean flip head tweezer. The pyrofoil was loaded onto a Curie-point pyrolyzer (JHS-3, Japan Analytical Industry Co., Ltd) coupled with a GC/MS (7890GC/5975MS. Agilent Technology) system, and was rapidly heated to 670°C at 5 s. The interface between Curie point and GC injection port was 300°C. The pyrolyzed compounds were separated with a DB-5ms capillary column (30 m × 0.25 mm × 1 µm film thickness; J&W Scientific, USA). The initial the GC oven temperature was at 50°C for 5 min, increase at a rate of 25°C min$^{-1}$ to 300 °C, and then hold at the final temperature of 310°C for 10 min. The carrier gas was ultra-high purity helium (He). The mass selective detector (5975, Agilent Technology) was at a full scan mode from 30 to 500 amu under electron ionization (EI) at a voltage of 70 eV and an ion source temperature of 230°C. Peaks were identified from the known fragmentation, mass spectra, and retention times of the target pyrolysis products of detected MPs (Table S1). Calibration curves were prepared using the reference standards. All standards excepting rubbers (99%, JSR Corporation) were purchased from Dupont (≥ 98%, USA). The linearities of the standard calibration curves were > 0.987. As markers for butadiene rubber (BR) (e.g., vinylcyclohexene) have interferences from SBR, BR is not quantified independently; instead, the sum of SBR and BR is reported to reduce uncertainties."*

Line 140 – please provide manufacturer information on the thermal desorption unit used.

Response:

The statement has been revised as follows.

*"An in-port thermal tube (78 mm long, 4 mm I.D., 6.35 mm O.D., Agilent Technology, USA) coupled with a GC/MS system (7890GC/5975MS, Agilent Technology, USA) was utilized to analyze phthalates."*

Line 142 (and elsewhere) – please provide supplier information and purity for all chemicals used in the study.

Response:

The supplier details and purity of all chemicals used in this study have been added to the Methods section.

*"An in-port thermal tube (78 mm long, 4 mm I.D., 6.35 mm O.D., Agilent Technology, USA) coupled with a GC/MS system (7890GC/5975MS, Agilent Technology, USA) was utilized to analyze phthalates, including dimethylphthalate (DMP), diethyl phthalate (DEP), di-n-butyl phthalate (DBP), butyl benzyl phthalate (BBP), bis(2-ethylhexyl)phthalate (DEHP), and di-n-octyl phthalate (DnOP) All PAEs were purchased from Sigma-Aldrich (≥ 98%, Steinheim, Germany)."*
*"Nine types of benzothiazole (97%, Thermofisher Scientific Co., LTD, Waltham, MA, United States) related compounds were quantified"*

Line 154 – the extraction and concentration procedures need to be provided in detail.

Response:

Thank you for your comments. The extraction and concentration procedures were performed as detailed in Appendix 2 in Supplemental Information, including ultrasonic extraction and purification.

*"The filter is transferred into a test tube, and 10 mL of a mixture of ultrapure deionized water (18 M-Ohm) and methanol (HPLC grade, Fisher Chemical, USA) (5:3, v:v) is added. The sample is extracted in an ultrasonic water bath at room temperature for 60 min. The combined extracts are concentrated and then diluted with ultrapure deionized water containing 0.2% v/v formic acid (~pH 2.5). The diluted extract is purified using an Oasis HLB Flangeless Vac Cartridge (3cc, 60 mg sorbent per cartridge, 30 μm particle size; Waters, USA)."*

Line 158 – please provide method details instead of referring to a different paper.

Response:

We sincerely appreciate your suggestion. As requested, we have now provided the method details in Supplementary Information (Appendix 2). The specific information is as follows:

*"**Appendix 2** Analysis of plasticizers*
*Phthalates were quantified using in-injection port-thermal desorption/mass spectrometry (TD-GC/MS) method. Aliquots of the filters (1.578 cm2) were cut into small pieces, spiked with ISs (Chrysene-d12, 96%, LGC Standard Limited, United States), and inserted into thermal tubes (78 mm long, 4 mm I.D., 6.35 mm O.D., Agilent Technology, USA) for analyses. The sample tube was directly loaded into a GC injection port (GC7890, Agilent Technology), at an initial temperature of 50°C. The temperature of injector was then ramped to 275°C for desorption in a splitless mode, while the GC oven temperature was kept at 30°C. The desorbed analytes were refocused at the column head. After the injector temperature reaches the set point, the oven program starts. The analytes were speared by an DB-5ms capillary column (30 m × 0.25 mm i.d. × 0.25 μm film*

*thickness; J&W Scientific). The carrier gas was ultra-high purity (99.9999%) He at a constant flow of 1.0 cm3 min-1. The MSD (5975, Agilent Technology) was full scanned from 50 to 550 amu under electron impact ionization (EI) at a voltage of 70 eV and an ion source temperature of 230°C. Identification was achieved by characteristic ion and retention times of the chromatographic peaks with those of authentic standards.*

*To quantify benzothiazole and its derivatives, each of 1.578 cm2 of the filter sample was cut and spiked with an internal standard (IS) of benzothiazole-d4 (benzothiazole=d4, 95%, LGC Standard Limited, United States). The filter is transferred into a test tube, and 10 mL of a mixture of ultrapure deionized water (18 M-Ohm) and methanol (HPLC grade, Fisher Chemical, USA) (5:3, v:v) was added. The sample was extracted in an ultrasonic water bath at room temperature for 60 min. The combined extracts were concentrated and then diluted to with ultrapure deionized water containing 0.2% v/v formic acid (~pH 2.5). The diluted extract was purified using an Oasis HLB Flangeless Vac Cartridge (3cc, 60 mg sorbent per cartridge, 30 μm particle size; Waters, USA). The target analytes were eluted with 5 mL of methanol and the eluents were evaporated to 1 mL under a gentle nitrogen stream prior to analysis. The separation of target analytes was performed using an ultra-performance liquid chromatography (UPLC; ACQUITY, Waters), and both identification and quantification were accomplished using a triple quadrupole mass spectrometer (ESI-MS/MS; Xevo TQ-S, Waters). An ACQUITY UPLC BEH SHIELD RP 18 column (100 mm ×3 mm × 1.7 mm) was serially connected to a Vanguard column (BEH C18, 5 mm × 2.1 mm × 1.7 mm). The mobile phase comprises 100% methanol (A) and ultrapure deionized water acidified with 0.1% (v/v) formic acid (B) at a flow rate of 450 mL min-1. A gradient elution program was applied for the separation. The tandem MS system was operated in the positive ion multiple reaction monitoring mode. Identification was achieved by characteristic ion and retention times of the chromatographic peaks with those of authentic standards."*

Line 167 – please discuss potential for interferences with detection or any specificity testing that was completed.

Response:

In order to avoid fluorescence detector saturation owing to matrix components in the quantification of BPA, the excitation and emission wavelengths from 0 to 14 min were fitted at 210 and 220 nm, respectively. Besides, to eliminate potential interferences from other organic compounds in PM samples, we employed both peak purity testing and spectrum matching in the quantification of BPA. For peak purity, we set a scan threshold of 1 mAU and required a peak coverage of 95%. For spectrum matching, we established a similarity threshold of 0.98. These rigorous measures effectively confirmed the presence of BPA in the samples, free from interference from other compounds.

Line 168 – was quantification completed using an internal standard or external standard method?
Response:

Quantification of total bisphenol A (BPA) in this study was completed using an external standard method. Calibration curves for BPA were prepared using BPA standard (≥ 99%, Sigma-Aldrich, Steinheim, Germany) solutions in acetonitrile. The linearities of the standard calibration curves were > 0.987. The recoveries of bisphenol A (BPA) are in a range of 96.3%-99.3%.

Section 2.3. – With regard to the FB filters, presumably many combustion products would be produced that no longer resemble the chemical structure of the original plastic. Please discuss the potential of these compounds to influence the oxidative testing.

Response:

Thank you for your valuable comment. Our results demonstrated that PMs from combustion sources exhibited higher oxidative potential (OP) compared to non-combustion sources. This suggests that the combustion process may enhance the OP of PM. This phenomenon may be associated with the products generated from the combustion of plastics. However, as we were unable to isolate these combustion products from PM, we could not quantify the impact of polymer combustion products on the OP results.

In our study, we focused on exploring the relationship between the DTT activity of PMs and microplastics and plasticizers using a correlation method. We focused on the oxidative potential of residual microplastics and plasticizers after combustion, rather than their thermal transformation products.

We acknowledge the importance of investigating the specific contributions of combustion-derived microplastics to PM OP. In future research, we propose conducting DTT assays on standard microplastic subjected to combustion treatment, comparing with non-combusted microplastics. This approach could provide valuable insights into the role of combustion products of microplastics in altering the oxidative potential of PM.

Line 221 – since wax coatings often contain a mixture of alkanes, please discuss any potential interferences of the wax on the pyrolysis GC/MS results (e.g. PE quantification).

Response:

The reason we distinguish between plastic burning (PB) and fruit bag burning (FB) rather than classifying them as a single combustion source is that the wax layer in fruit bags cannot be separated from the plastic. This is a featured source in the Guanzhong region (This is also quite common in fruit producing areas in northern China), and local residents typically burn fruit bags directly without separating the wax.

By comparing the results of PB and FB, we found that, there were no significant differences in the composition and concentration of microplastics and plasticizers emitted from PB and FB. These results indicate that the wax coating in FB has minimal influence on the analytical results.

Line 223 – the MPs in PB and LB were 50% of the PM, was this by mass? Please clarify and also present the percentage of PM that was attributed to plastic in all scenarios.

Response:

We apologize for causing confusion to the reviewer. The 50% mentioned in original Line 223 refers to the relative proportion of coarse and fine microplastics from PB and LB, meaning that the combined mass percentage of coarse and fine microplastics is 100%. The sentence has been revised as follows.

*"Notably, MPs in Plastic Burning and Livestock Breeding constituted a comparable proportion in both fine and coarse fractions, both close to 50%."*

We also presented the mass proportion of microplastics in PMs from five sources in Figure S4.

[Figure]

**Figure S4** The mass proportion of MPs in PMs from five sources (PB: Plastic Burning, FB: Fruit-bag Burning, RT: Road Traffic, AF: Agricultural Film, LB: Livestock Breeding).

Line 226 – how was the reported uncertainty calculated?

Response:

At each sampling location, we collected 2-3 replicate samples for each particle size fraction. The standard deviation was calculated to represent the uncertainty.

Line 261 – Please discuss the details of the flyover? How was this sample collection done?

Response:

The sampling was conducted on a pedestrian flyover (height: ~4 m) near the Xi'an Jiaotong University campus in Beilin District, Xi'an. The flyover spans an urban arterial road with four motorized lanes (bidirectional), two dedicated bicycle lanes. The traffic volume is substantial, with private cars being the primary type of vehicle. We did not choose to conduct sampling on the ground by the roadside in order to avoid direct interference from road dust and vehicle exhaust.

The MiniVOL samplers (Airmetrics, Springfield, OR, USA), inertial impactors, was conducted on the flyover railing (see the following photo). In order to prevent any potential interference from pedestrians on the flyover, we cordoned off an area of 1.5 meters around the sampler with caution tape and affixed signs indicating that the area was designated for scientific research purposes.

[Figure]

Line 394 – please cite where these compounds are accepted as carcinogens.

Response:

We have revised the sentence to include the necessary citations to support this statement.

*"ILCR for the three carcinogenic compounds (BT, BBP, and DEHP) were calculated in this study (Guyton et al., 2009; Ma et al., 2020; Liu et al., 2023)."*

Reference:

Guyton, K. Z., Chiu, W. A., Bateson, T. F., Jinot, J., Scott, C. S., Brown, R. C., and Caldwell, J. C.: A Reexamination of the PPAR-α Activation Mode of Action as a Basis for Assessing Human Cancer Risks of Environmental Contaminants, Environ. Health Perspect., 117, 1664-1672, 10.1289/ehp.0900758, 2009.

Ma, B. B., Wang, L. J., Tao, W. D., Liu, M. M., Zhang, P. Q., Zhang, S. W., Li, X. P., and Lu, X. W.: Phthalate esters in atmospheric PM2.5 and PM10 in the semi-arid city of Xi'an, Northwest China: Pollution characteristics, sources, health risks, and relationships with meteorological factors, Chemosphere, 242, 10, 10.1016/j.chemosphere.2019.125226, 2020.

Liu, M. X., Xu, H. M., Feng, R., Gu, Y. X., Bai, Y. L., Zhang, N. N., Wang, Q. Y., Ho, S. S. H., Qu, L. L., Shen, Z. X., and Cao, J. J.: Chemical composition and potential health risks of tire and road wear microplastics from light-duty vehicles in an urban tunnel in China, Environmental Pollution, 330, 9, 10.1016/j.envpol.2023.121835, 2023.

Line 432 – does correlation in this testing confirm causation of oxidative stress? Please discuss.

Response:

The strong correlation between these components (e.g. PET, PAEs, etc.) and DTT consumption rate only indicates that these components can promote DTT activities, suggesting their potential contribution to the oxidative potential (OP) of particulate matter. However, these results do not conclusively prove that these substances directly cause oxidative stress. Although OP exhibits a certain correlation with ROS generation, there is still a scarcity of mechanistic understanding in this area (Hwang et al., 2021; Zhang et al., 2024). We will also pay attention to this issue and conduct relevant research in the future.

Reference:

Hwang, B., Fang, T., Pham, R., Wei, J. L., Gronstal, S., Lopez, B., Frederickson, C., Galeazzo, T., Wang, X. L., Jung, H., and Shiraiwa, M.: Environmentally Persistent Free Radicals, Reactive Oxygen Species Generation, and Oxidative Potential of Highway PM$_{2.5}$, ACS Earth Space Chem., 5, 1865-1875, 10.1021/acsearthspacechem.1c00135, 2021.

Zhang, X. J., Wang, Y. D., Yao, K. X., Zheng, H., and Guo, H. B.: Oxidative potential, environmentally persistent free radicals and reactive oxygen species of size-resolved ambient particles near highways, Environ. Pollut., 341, 9, 10.1016/j.envpol.2023.122858, 2024.

---

## Author Comment (AC6)

**Reviewer 6:**

This paper investigates emitted microplastics aerosols from five sources in northwest China. The authors present a comprehensive characterization of the sources and plasticizer profiles. An eco-health risk assessment was conducted, providing key details like daily exposure risks. These findings are especially valuable for informing risk-mitigation policies and protecting individuals who are often close to sources like those in rural households which burn plastics, agricultural workers or incineration plant workers. This paper merits publication in ACP after some minor revisions.

Dear reviewer,

Thanks very much for taking your time to review this manuscript. We have carefully considered each of your suggestions and have made the adequate revisions to improve the quality of our manuscript.

General comments:

1. The current manuscript misses some clarification and details on the different source categories. For example, it only became apparent in Table 1 that the plastic burning occurred in a household setting not in an industrial one. This distinction is important, as emissions from household and industrial plastic burning may differ significantly, and readers should not conflate the two. Thus, this point should be clarified and emphasized in the Introduction. While it appears that fruit bags were not included in the household plastic burning, it would be helpful to define 'fruit bags' more thoroughly. What polymer types are they made out of - line 105 only states 'also containing some plastics components'. How do they differ from common household plastic waste? Clarifying this will help the readers to better understand why you chose the two categories 'fruit bag' vs 'plastic burning' and their differences.

Response:

Thank you for the reviewer's insightful comments. We have now explicitly stated that our study focused on life pollution sources and the plastic burning occurred in a household setting in the Introduction and Abstract sections. The related statements are as follows.

*"Current research on atmospheric MPs sources focuses on industrial emissions and natural processes, but neglects air pollution sources closely related to daily life sources. Given that living sources significantly affect human health, this study pays particular attention to such sources."*

*"Due to the relatively limited facilities and means of waste disposal, residents in rural areas often resort to open burning when disposing of plastic waste (Pathak et al., 2023). In addition, given the flammability of plastics, residents also tend to use plastics as igniters or even burn them directly when using stoves for cooking or heating, which is an important household source of MPs."*

*"The aims of this research are to characterize the distributions of MPs and plasticizers in dual-size PMs (PM$_{2.5}$, PM$_{2.5-10}$) from typical MP sources (anthropogenic sources from daily life) in the Guanzhong Plain."*

The reason we distinguish between plastic burning (PB) and fruit bag burning (FB) rather than classifying them as a single combustion source is that the wax layer in fruit bags cannot be separated from the plastic. This is a featured source in the Guanzhong region (This is also quite common in fruit producing areas in northern China), and local residents typically burn fruit bags directly without separating the wax. We have revised this point in the manuscript as follows.

*"The Guanzhong Plain is an important fruit production base in China, with the highest*

*consumption of fruit bags. Local residents often use the above-mentioned plastic products to ignite solid fuels for indoor heating or cooking. The reason we distinguish between Plastic Burning and Fruit-bag Burning rather than classifying them as a single combustion source is that the wax layer in fruit bags cannot be separated from the plastic. This is a featured source in the Guanzhong region (This is also quite common in fruit producing areas in northern China), and local residents typically burn fruit bags directly without separating the wax. Table 1 provides a summary of the essential details for each source."*

The more details of fruit bags and their differences with household plastic waste have been added in the manuscript as follows.

*"Fruit bags are typically lightweight, thin-film, which are designed for single-use and are often discarded after a short period, differing from common household plastic waste. These bags are usually made from low-density polyethylene and Nylon, which is known for its flexibility and transparency (Ali et al., 2021; Yang et al., 2022)."*

Reference:

Ali, M. M., Anwar, R., Yousef, A. F., Li, B. Q., Luvisi, A., De Bellis, L., Aprile, A., and Chen, F. X.: Influence of Bagging on the Development and Quality of Fruits, Plants-Basel, 10, 16, 10.3390/plants10020358, 2021.

Yang, H. G., Gu, F. W., Wu, F., Wang, B. K., Shi, L. L., and Hu, Z. C.: Production, Use and Recycling of Fruit Cultivating Bags in China, Sustainability, 14, 18, 10.3390/su142114144, 2022.

Pathak, G., Nichter, M., Hardon, A., Moyer, E., Latkar, A., Simbaya, J., Pakasi, D., Taqueban, E., and Love, J.: Plastic pollution and the open burning of plastic wastes, Glob. Environ. Change-Human Policy Dimens., 80, 9, 10.1016/j.gloenvcha.2023.102648, 2023.

2.  Details on the collection of field blanks are limited and would benefit from further clarification. Line 118: How were the field blanks collected? By synchronously, do the authors mean that the blank was taken at the same location, same duration, and same time as the samples?

Response:

We have provided the information on how the field blanks collected in Line 163-166.

*"The field blank of each type of source was synchronously collected with active sampling. Unused filters (the same batch as sampling filters) were loaded into identical sampling devices, which were placed adjacent to operational samplers for the entire duration of one sampling event."*

3.  Line 128: Why were phthalates, benzothiazole and its derivatives, and bisphenol A chosen to be the 3 classes for in-depth analysis? What criteria guided this choice?

Response:

The selection of BTs, PAEs, and BPA as the focus of our study is driven by their widespread use, ubiquitous in the environment and potential health risks.

[revised manuscript text omitted]

4. Near the source you may have metals and other aerosols depositing onto the filter, were these non-plastic sources accounted for in your analysis? How was the mass contribution from microplastics determined?

Response:

In this study, we employed Mass Spectrometry methods (Py-GCMS) for the identification and

quantification of microplastics and plasticizers. These methods detect marker ions unique to microplastics (MPs) and plasticizers, while metals and other aerosols do not produce these fragments (Hermabessiere et al., 2018).

The mass contribution from microplastics was determined by calculating the mass proportion of each type of MP to the total nine detected MPs in the samples.


Response:

Thank you for your suggestion. We have revised the manuscript to remove abbreviations of PB, FB, RT, AF, LB, and OP in the main text to enhance readability. However, in order to maintain conciseness and adhere to the word limit, we have retained the abbreviations in the Abstract and Figures/Tables.

Specific comments:

■  Line 89: Can you elaborate why is the UV so strong in this area and why does it matter for this study? MPs emitted in this area may undergo increased photooxidation, is that something that can be discussed with your results? Or is that something that needs to be explored further in future studies?

Response:

The Guanzhong Plain, our study area is located in northwestern China, and according to the study "Ultraviolet radiation over China: Spatial distribution and trends (Sust. Energ. Rev., 76, 1371-1383)", this region experiences relatively strong UV radiation (14486 kJ $m^{-2}$ $day^{-1}$, the average solar radiation of Xi'an, Tongchuan, and Xianyang). The high UV radiation in this area can be attributed to the elevated altitude (497 m, the average elevated altitude of Xi'an, Tongchuan, and Xianyang) (reducing atmospheric path length) and less cloud cover (especially low cloud cover). Ultraviolet radiation drives photooxidation of plastics, breaking polymer chains and accelerating fragmentation into secondary MPs. The high UV intensity in the Guanzhong Plain exacerbates this process.

However, our current research primarily focused on the characteristics and health risks of MPs and plasticizers emitted from various sources in the region. We did not specifically examine how UV-induced aging of MPs might influence their environmental behavior or toxicity. This aspect is important as ultraviolet radiation can alter the physical and chemical properties of MPs, potentially increasing their toxicity and capacity to adsorb other pollutants. We accept the reviewer's suggestion and plan to explore the effects of UV on MPs and their associated health

risks in future studies. This will provide a more comprehensive understanding of the eco-health impacts of MPs in the Guanzhong Plain.

Reference:

Liu, H., Hu, B., Zhang, L., Zhao, X. J., Shang, K. Z., Wang, Y. S., and Wang, J.: Ultraviolet radiation over China: Spatial distribution and trends, Renew. Sust. Energ. Rev., 76, 1371-1383, 10.1016/j.rser.2017.03.102, 2017.

■ Line 113: State what type of sampler MiniVol samplers are (e.g. impinger, impactor, etc.)

Response:

Thank you for this comment. The MiniVol air samplers used in this study are impactors. This point has been revised as follows.

*"MiniVOL samplers (Airmetrics, Springfield, OR, USA), which are inertial impactors, were employed for collection, operating at a steady flow rate of 5 L min$^{-1}$."*

■ Figure 1: errors bars used represent standard deviation of PM10 however the figure is plotting PM5 and PMcourse. Is that error representation suitable for this plot? Consider just focusing on PM2.5 and PMcourse throughout the paper and removing PM10 to be more consistent with how results are discussed. Is that error representation suitable for this plot? Consider just focusing on PM2.5 and PMcourse throughout the paper and removing PM10 to be more consistent with how results are discussed.

Response:

We have revised Figure 1 to illustrate the characteristics of microplastics and plasticizers in PM$_{2.5}$ and PM$_{2.5-10}$ (PM$_{coarse}$) and remove PM$_{10}$ from this figure. Additionally, we have included error bars representing the standard deviation for both PM$_{2.5}$ and PM$_{2.5-10}$.

[Figure]

**Figure 1** Average concentrations of MPs (a) and plasticizers (b) in PM$_{2.5}$ and PM$_{2.5-10}$ from five sources (PB: Plastic Burning, FB: Fruit-bag Burning, RT: Road Traffic, AF: Agricultural Film, LB: Livestock Breeding).

■ Figure 1: I would suggest increasing the spacing between (a) and (b) to improve readability. The secondary y-axis for (a) is very close the first y-axis of (b).

Response:

Suggestion taken. The figure was shown as above.

■ Line 282: In addition to the most abundant plasticizer, I would suggest the authors list the next two most abundant plasticizer types detected in PB.

We apologize for causing confusion to the reviewer. It has been revised as follows.

*"The highest concentrations of PAEs, BTs, and BPA still appear in PB among five sources."*

■ Line 288-289 and Figure 5: Use consistent spelling of tire or tyre throughout the paper.

Response:

Sorry for the error. We have revised line 288-289 and Figure 5 to consistently use the spelling "tire" throughout the revised manuscript.

■ Figure 3: Redefine abbreviations when they appear in captions e.g., PAE and BT

Response:

Suggestion taken.

■ Figure 4: Consider adding visual cues (e.g. labels by the arrow, highlighted section or color cues) to help guide the reader's eye from the black arrow to the corresponding source marker on the x-axis. In the main text, please expand the discussion for these identified markers. How confident are these marker classifications? Have previous studies reported the same markers?

Response:

We have added the abbreviation of the fingerprint species next to the arrow in Figure 4 to highlight the substance. The revised Figure is as follows.

[Figure]

**Figure 4** Source profiles of microplastics and plasticizers in $PM_{2.5}$ and $PM_{10}$ (The black arrows indicate the source markers).

The marker classifications in this study were based on the distinct emission characteristics of different sources. We compared and selected these markers to represent each source accurately based on our real-world experiment results. While previous studies on the emission characteristics of microplastics from these sources are relatively limited, there are some findings that corroborate our results. We have expanded the discussion for the identified markers in the manuscript as follows.

"Liu et al. (2019) documented that polystyrene (PS) were the predominant polymer in agricultural sources."

"Liu et al. (2023) revealed that natural rubber (NR) and other rubber particles are emitted at high levels in tunnel traffic, emerging as the dominant microplastic in traffic-dominated environments."

■ Figure 6: Please add a legend to define the colour groupings.

Response:

Suggestion taken.

[Figure]

**Figure 6** Correlations between DTT, MPs, and plasticizers (*P < 0.05; **P < 0.01).

Technical corrections:

■ Line 19: (phthalates, benzothiazole and its derivatives, and bisphenol A)

Response:

Suggestion taken. It has been revised as follows.

"the characteristics and source profiles of eight types of common MPs and three classes of plasticizers (i.e., phthalates, benzothiazole and its derivatives, and bisphenol A)."

■ Line 31: missing closing bracket: poly(methyl methacrylate)

Response:

Corrected.

■ Line 122: missing space: …frozen at -20°C

Response:

Corrected.

■ Line 180: and absorbance was measured at 412 nm using a microplate reader

Response:

Corrected.

- Line 246: This is because plastic products...

Response:

Corrected.

---

## Author Response (AR1)

**Reviewer 1:**

Focusing on atmospheric microplastic pollution in the Guanzhong Plain of China, this paper investigates the characteristics of microplastics and plasticizers emitted from five typical sources of microplastics (plastic burning, fruit bag burning, road traffic, agricultural films, and animal husbandry) and their ecological health impacts, to provide a comprehensive perspective for a deeper understanding of atmospheric microplastic pollution. It is recommended that this manuscript be published with minor revisions.

Dear reviewer,

Thanks very much for taking your time to review this manuscript. We have carefully considered each of your suggestions and have made the necessary revisions to improve the quality of our manuscript. Below are our detailed responses to your comments:

1. Line 152: OBS and CBS all appear for the first time in the abstract, and it is recommended that the abbreviations be labelled on the first occurrence.

**Response:**

We have labeled the abbreviations in the abstract as requested. Additionally, we have ensured that these abbreviations are again defined at their first occurrence in the main text.

2. Line 175: note the font of the concentration units.

**Response:**

We have reviewed the concentration units and ensured that they are consistently formatted in the same font style throughout the manuscript.

3. Why choose BT, PAE, and BPA as plastic additives for in-depth research in this study? Response:

The selection of BTs, PAEs, and BPA as the focus of our study is driven by their widespread use, ubiquitous in the environment and potential health risks.

Phthalate esters (PAEs) are the most widely used plasticizers globally, dominating the plastic additive market. He et al. (2020) demonstrated that during 2007-2017, the annual global production of PAEs increased from 2.7 million tons to 6 million tons. Moreover, China is recognized as the largest importer of PAEs worldwide. Benzothiazoles (BTs) are extensively used in automotive tires and agrochemicals. High concentrations of BTs were discovered in the street runoff, suggesting that these tire material-related compounds can persevere in the environment (Zhang et al., 2018). Exposure to BTs may result in central nervous system depression, liver and kidney damage, dermatitis, and pulmonary irritation (Ginsberg et al., 2011). Bisphenol A (BPA) as a common industrial chemical component in many products, has steadily grown over the last 50 years (Corrales et al., 2015). Growth of global production has consistently ranged between 0% and 5% annually (Corrales et al., 2015). PAEs and BPA considered as endocrine disruptors, are demonstrated to impair reproductive function and development in laboratory animals (Wang et al., 2018).

Despite the well-documented health risks associated with these plasticizers in laboratory settings, there is a significant gap in understanding their real-world emissions and health impacts. In this study, we aim to fill this gap by investigating the emission characteristics of these plasticizers from various sources and evaluating their potential health impacts based on real-world

concentration levels.

The related sentences have been revised as follows.

"Plasticizers are widely used in the production of plastics in order to achieve the desired material properties (Demir and Ulutan, 2013). Since plasticizers are not chemically bound to the plastic products, they can easily diffuse into the surrounding environment during the life-time (Yadav et al., 2017; Demir and Ulutan, 2013). PAEs, BTs, and BPA are the most common plastic additives that are ubiquitous in the environment and pose potential health risks. Phthalate esters (PAEs) are the most widely used plasticizers globally, dominating the plastic additive market. He et al. (2020) demonstrated that during 2007-2017, the annual global production of PAEs increased from 2.7 million tons to 6 million tons. China is recognized as the largest importer of PAEs worldwide. Benzothiazoles (BTs) are extensively used in automotive tires and agrochemicals. High concentrations of BTs were discovered in the street runoff, suggesting that these tire material-related compounds can persevere in the environment (Zhang et al., 2018). Exposure to BTs may result in central nervous system depression, liver and kidney damage, dermatitis, and pulmonary irritation (Ginsberg et al., 2011). Bisphenol A (BPA) as a common industrial chemical component in many products, has steadily grown over the last 50 years (Corrales et al., 2015). Growth of global production has consistently ranged between 0% and 5% annually (Corrales et al., 2015). PAEs and BPA considered as endocrine disruptors, are demonstrated to impair reproductive function and development in laboratory animals (Wang et al., 2019).

Previous studies have investigated the emission characteristics of plasticizers from various sources. Simoneit et al. (2005) illustrated that the major plasticizers detected in particulate matters (PMs) from open-burning of plastics were dibutyl phthalate (DBP), diethylhexyl adipate (DEHA), and diethylhexyl phthalate (DEHP). Zeng et al. (2020) reported phthalate concentrations in greenhouses air were higher than that in ambient air. Liu et al. (2023) found that phthalates were the most dominant plasticizer compositions in tunnel PM2.5, accounting for 64.8% of the detected plasticizers. Zhang et al. (2018) demonstrated that tire material-related compounds, benzothiazole (BT) and 2-hydroxybenzothiazole (2-OH-BT) were the major compounds in both tire and road dust samples. The majority of existing studies on atmospheric MPs and plasticizers have focused on analyzing the emission characteristics of individual source and lacked a comprehensive and comparative analysis of the MPs emission profiles of various sources."

**Reference:**

- Corrales, J., Kristofco, L. A., Steele, W. B., Yates, B. S., Breed, C. S., Williams, E. S., and Brooks, B. W.: Global Assessment of Bisphenol A in the Environment: Review and Analysis of Its Occurrence and Bioaccumulation, Dose-Response, 13, 29, 10.1177/1559325815598308, 2015.
- Demir, A. P. T. and Ulutan, S.: Migration of phthalate and non-phthalate plasticizers out of plasticized PVC films into air, Journal of Applied Polymer Science, 128, 1948-1961, 10.1002/app.38291, 2013.
- Ginsberg, G., Toal, B., and Kurland, T.: Benzothiazole Toxicity Assessment In Support Of Synthetic Turf Field Human Health Risk Assessment, J. Toxicol. Env. Health Part A, 74, 1175-1183, 10.1080/15287394.2011.586943, 2011.
- He, M. J., Lu, J. F., Wang, J., Wei, S. Q., and Hageman, K. J.: Phthalate esters in biota, air and

- water in an agricultural area of western China, with emphasis on bioaccumulation and human exposure, Sci. Total Environ., 698, 9, 10.1016/j.scitotenv.2019.134264, 2020.
- Liu, M. X., Xu, H. M., Feng, R., Gu, Y. X., Bai, Y. L., Zhang, N. N., Wang, Q. Y., Ho, S. S. H., Qu, L. L., Shen, Z. X., and Cao, J. J.: Chemical composition and potential health risks of tire and road wear microplastics from light-duty vehicles in an urban tunnel in China, Environmental Pollution, 330, 9, 10.1016/j.envpol.2023.121835, 2023.
- Simoneit, B. R. T., Medeiros, P. M., and Didyk, B. M.: Combustion products of plastics as indicators for refuse burning in the atmosphere, Environ. Sci. Technol., 39, 6961-6970, 10.1021/es050767x, 2005.
- Wang, Y., Zhu, H. K., and Kannan, K.: A Review of Biomonitoring of Phthalate Exposures, Toxics, 7, 28, 10.3390/toxics7020021, 2019.
- Yadav, I. C., Devi, N. L., Zhong, G., Li, J., Zhang, G., and Covaci, A.: Occurrence and fate of organophosphate ester flame retardants and plasticizers in indoor air and dust of Nepal: Implication for human exposure, Environmental Pollution, 229, 668-678, 10.1016/j.envpol.2017.06.089, 2017.
- Zeng, L.-J., Huang, Y.-H., Chen, X.-T., Chen, X.-H., Mo, C.-H., Feng, Y.-X., Lu, H., Xiang, L., Li, Y.-W., Li, H., Cai, Q.-Y., and Wong, M.-H.: Prevalent phthalates in air-soil-vegetable systems of plastic greenhouses in a subtropical city and health risk assessments, Sci. Total Environ., 743, 10.1016/j.scitotenv.2020.140755, 2020.
- Zhang, J., Zhang, X., Wu, L., Wang, T., Zhao, J., Zhang, Y., Men, Z., and Mao, H.: Occurrence of benzothiazole and its derivates in tire wear, road dust, and roadside soil, Chemosphere, 201, 310-317, 10.1016/j.chemosphere.2018.03.007, 2018.
- Zhang, H., Yang, R. F., Shi, W. Y., Zhou, X., and Sun, S. J.: The association between bisphenol A exposure and oxidative damage in rats/mice: A systematic review and meta-analysis, Environmental Pollution, 292, 9, 10.1016/j.envpol.2021.118444, 2022.
- 4. Please supplement QAQC in the Methods section for the analysis of DTT, need to know the accuracy and precision of the analysis.

Thank you for your suggestion. We have supplemented the Methods section with the following detailed information:

- "To ensure the accuracy of the results, the entire experiment was performed under dark conditions. Prior to sample analysis, a standard curve was generated by measuring the absorbance of 11 DTT concentration gradients within the range of 0 to 450  $\mu$ mol  $L^{-1}$ , achieving a correlation coefficient ( $R^2$ ) of 0.9997. Pure methanol solution was used as a blank control, which was processed and measured in the same manner as the samples. The DTT consumption rate of each sample was corrected using the DTT consumption rate of the blank. Each batch of samples and methanol blanks was measured in duplicate to verify experimental reproducibility. The linear fitting  $R^2$  for DTT consumption rates was consistently greater than 0.9, and the coefficient of variation (standard deviation) for parallel experiments was less than 15%."
- 5. How is the Source Pathway Decoder model developed? Is there an operation based on some data? Input of basic information? Or is it just a conceptual model? What is the main purpose in this study?

The Source Pathway Receptor model was developed based on methodologies described in the systematic review by Velis et al. (2021) titled "Mismanagement of Plastic Waste through Open Burning with Emphasis on the Global South: A Systematic Review of Risks to Occupational and Public Health". It is just a conceptual model. This model systematically evaluates the environmental consequences of MPs and plasticizers from different sources and their inter-relationships. Based on the emission concentrations and potential pathways of MPs and plasticizers from typical sources, studies can be conducted to assess the effects of MPs and plasticizers to human health at environmentally relevant concentrations in the future.

**Reference:**

Velis, C. A. and Cook, E.: Mismanagement of Plastic Waste through Open Burning with Emphasis on the Global South: A Systematic Review of Risks to Occupational and Public Health, Environ. Sci. Technol., 55, 7186-7207, 10.1021/acs.est.0c08536, 2021.

6. Lines 210 and 429: It is suggested that there should be consistency throughout the text as to whether P should be capitalized or lower case.

**Response:**

We have revised the text to ensure consistent capitalization by using uppercase "P" throughout the manuscript.

7. Line 296: It is recommended that further discussion of the reasons why the results in this study are contrary to other studies.

**Response:**

Thank you for your suggestion. We have added a more detailed discussion in the revised manuscript.

"Furthermore, BTs, PAEs, and BPA from sources except for PB were prevalent in PM2.5 relative to PM2.5-10, contrary to the results reported by Nunez et al. (2020). This discrepancy may be attributed to differences in pollution sources. Nunez et al. (2020) demonstrated that port industrial activities (e.g., cargo handling and industrial emissions) predominantly generated coarse PM, resulting in higher concentrations of plasticizers in this fraction. In contrast, high temperature in PB promoted the formation of fine particles with larger surface area enhancing the adsorption of plasticizers."

**Reference:**

Nunez, A., Vallecillos, L., Maria Marce, R., and Borrull, F.: Occurrence and risk assessment of benzothiazole, benzotriazole and benzenesulfonamide derivatives in airborne particulate matter from an industrial area in Spain, Sci. Total Environ., 708, 10.1016/j.scitotenv.2019.135065, 2020.

8. Line 438: It is recommended that an outlook for future research be included. For example, there may be a variety of other MPs in the environment that were not detected in this study, which may pose a risk, and it is recommended that the need to expand the range of substances assessed be mentioned in the outlook for future studies to improve the health risk assessment system.

We sincerely appreciate the reviewer's valuable suggestion. We have revised the conclusion section to provide a more comprehensive outlook for future studies.

"Future studies should expand the range of assessed MPs and plasticizers and integrating multiple ecological health assessment methods to further refine the health risk assessment system and deepen the understanding of the environmental and health hazards of MPs."

9. Line 584: note the format of references.

**Response:**

The reference format has been carefully checked and revised to ensure consistency with the journal's guidelines.

10. You can add this newly reference: The hidden threat of microplastics in the bloodstream. The Innovation Life 3:10013

**Response:**

The authors have added the reference in Line 106 of revised manuscript. This reference elucidates the potential health risks of MPs when they enter the human bloodstream. These findings support our discussion on the potential health hazards of microplastics in our research.

**Reviewer 2:**

This paper investigates emitted microplastics aerosols from five sources in northwest China. The authors present a comprehensive characterization of the sources and plasticizer profiles. An eco-health risk assessment was conducted, providing key details like daily exposure risks. These findings are especially valuable for informing risk-mitigation policies and protecting individuals who are often close to sources like those in rural households which burn plastics, agricultural workers or incineration plant workers. This paper merits publication in ACP after some minor revisions.

**Response:**

We sincerely appreciate the time and effort you have dedicated to reviewing our manuscript. We also thank you for your meticulous attention to detail, which has helped us improve the clarity and accuracy of the manuscript. We have carefully addressed each suggestion and implemented the necessary revisions. Below are our detailed responses to your comments:

**General comments**

The authors must be aware of using past and present tense. Many places are inconsistent with using the correct tense.

**Response:**

Thank you for your suggestion. We have conducted a systematic review of verb tenses throughout the manuscript. Specifically, we have used the past tense throughout the sections on experimental methods and results to accurately reflect completed actions. In the discussion section, we have employed the present tense to describe current interpretations and implications of the findings.

1. Line 19: Add "i.e.," in front of phthalates

Response:

Suggestion taken.

2. Add a comma after (AF).

Response:

Suggestion taken.

3. Replace the verb "features".

**Response:**

This point has been revised as follows.

"PB exhibits high proportions of poly(methyl methacrylate) (PMMA) and 2-hydroxy benzothiazole (HOBT), with PMMA being more abundant in coarse particles (PMcoarse)."

4. Line 36: Spell out ROS to keep consistency.

Response:

Suggestion taken.

5. Line 43: in a range of "five millimeters to one micrometer"

Response:

Suggestion taken.

6. Line 53: It is recommended to change "results" to "result" because the subject is plural.

**Response:**

Suggestion taken.

**7. Line 59: Define PMs.**

**Response:**

The definition of PMs has been added.

"Evangeliou et al. (2020) have estimated that annual total global tire wear particle emissions were 2907 kt  $y^{-1}$ , with 29 and 288 kt  $y^{-1}$  for  $PM_{2.5}$  (particulate matter with aerodynamic diameters  $\leq 2.5 \ \mu m$ ) and  $PM_{10}$  ( $\leq 10 \ \mu m$ ), respectively."


**Response:**

Suggestion taken. The revised Table 1 is as follows:

Table 1 Basic sampling information of target emission sources

| Emission        | Sampling     | Sampling   | Sample | Sampling location                 |
|-----------------|--------------|------------|--------|-----------------------------------|
| source          | duration (h) | height     | No.    |                                   |
| Plastic burning | 2.0          | 3-4 m      | 5      | Open space, about 1 m             |
| (PB)            |              | above the  |        | downwind of chimney of rural      |
|                 |              | ground     |        | household stove in rural          |
| Fruit bag       | 2.0          | 3-4 m      | 5      | Xianyang                          |
| burning         |              | above the  |        |                                   |
| (FB)            |              | ground     |        |                                   |
| Road Traffic    | 13.6-14.1    | 3 m above  | 5      | Open space, flyovers on traffic   |
| (RT)            |              | the ground |        | arteries in downtown Xi'an        |
| Agricultural    | 24           | 1.5 m      | 5      | Open space, about 2 m away        |
| film            |              | above the  |        | from the greenhouse in farmland   |
| (AF)            |              | ground     |        | in rural Tongchuan                |
| Livestock       | 2.5-3.5      | 1.5 m      | 5      | About 1 m from the feed trough    |
| breeding        |              | above the  |        | in a cow shed of approximately 8  |
| (LB)            |              | ground     |        | m 2 in rural Tongchuan |

15. Line 122" A missing space for 'at -20°C'.

**Response:**

Suggestion taken.

16. Figure 1: It is recommended that the error bars should be thickened to increase the clarity. Response:

Suggestion taken. The modified Figure 1 is as follows:

Figure 1 Average concentrations of MPs (a) and plasticizers (b) in PM2.5 and PM2.5-10 from five

**Section 2.2.**

17. Line 147: More advanced information on the procedures of the GC/MS analysis must be shown (i.e. capillary column used, and MSD setting), not just referring to a reference.

Response:

We sincerely appreciate your suggestion. As requested, we have now provided the detailed GC/MS analytical procedures, including capillary column, MSD settings, and other relevant parameters, in Supplementary Information (Appendix 2). The specific information is as follows: "Appendix 2 Analysis of plasticizers

Phthalates were quantified using in-injection port-thermal desorption/mass spectrometry (TD-GC/MS) method. Aliquots of the filters (1.578 cm²) were cut into small pieces, spiked with ISs (Chrysene-d12, 96%, LGC Standard Limited, United States), and inserted into thermal tubes (78 mm long, 4 mm I.D., 6.35 mm O.D., Agilent Technology, USA) for analyses. The sample tube was directly loaded into a GC injection port (GC7890, Agilent Technology), at an initial temperature of 50°C. The temperature of injector was then ramped to 275°C for desorption in a splitless mode, while the GC oven temperature was kept at 30°C. The desorbed analytes were refocused at the column head. After the injector temperature reaches the set point, the oven program starts. The analytes were speared by an DB-5ms capillary column (30 m  $\times$  0.25 mm i.d.  $\times$  0.25 µm film thickness; J&W Scientific). The carrier gas was ultra-high purity (99.9999%) He at a constant flow of 1.0 cm³ min-1. The MSD (5975, Agilent Technology) was full scanned from 50 to 550 amu under electron impact ionization (EI) at a voltage of 70 eV and an ion source temperature of 230°C. Identification was achieved by characteristic ion and retention times of the chromatographic peaks with those of authentic standards.

To quantify benzothiazole and its derivatives, each of 1.578 cm2 of the filter sample was cut and spiked with an internal standard (IS) of benzothiazole-d4 (benzothiazole=d4, 95%, LGC Standard Limited, United States). The filter is transferred into a test tube, and 10 mL of a mixture of ultrapure deionized water (18 M-Ohm) and methanol (HPLC grade, Fisher Chemical, USA) (5:3, v:v) was added. The sample was extracted in an ultrasonic water bath at room temperature for 60 min. The combined extracts were concentrated and then diluted to with ultrapure deionized water containing 0.2% v/v formic acid (~pH 2.5). The diluted extract was purified using an Oasis HLB Flangeless Vac Cartridge (3cc, 60 mg sorbent per cartridge, 30 µm particle size; Waters, USA). The target analytes were eluted with 5 mL of methanol and the eluents were evaporated to 1 mL under a gentle nitrogen stream prior to analysis. The separation of target analytes was performed using an ultra-performance liquid chromatography (UPLC; ACQUITY, Waters), and both identification and quantification were accomplished using a triple quadrupole mass spectrometer (ESI-MS/MS; Xevo TQ-S, Waters). An ACQUITY UPLC BEH SHIELD RP 18 column (100 mm ×3 mm × 1.7 mm) was serially connected to a Vanguard column (BEH C18, 5 mm × 2.1 mm × 1.7 mm). The mobile phase comprises 100% methanol (A) and ultrapure deionized water acidified with 0.1% (v/v) formic acid (B) at a flow rate of 450 mL min-1. A gradient elution program was applied for the separation. The tandem MS system was operated in the positive ion multiple reaction monitoring mode. Identification was achieved by characteristic ion and retention times of the chromatographic peaks with those of authentic standards."

18. More detailed QC/QA procedures must be given for all analyses.

**Response:**

Thank you for your suggestion. The details of QC/QA procedures have already been documented in Section 2.3 of the Methods and as follows.

"Quality assurance/Quality control (QA/QC)

The flow rates of all samplers were calibrated using a mass flowmeter (Model 4140, TSI, Shoreview, MN, USA) before and after each sampling cycle. All quartz filters used in this study were preheated at 800°C for 3 h to remove any potential contaminants and then cooled before use. To minimize experimental error, sampling was conducted in duplicate for each particle size of each source. For the chemical measurements, one in every 10 samples was reanalyzed for quantity assurance purposes, and the SD errors of replicate trials were within 10% for the pyrolysis analyses. Calibration curves were established using reference standards. The linearities of the standard calibration curves were > 0.987. The standard deviations of the pyrolyzed standard were within 94.1% to 98.3%. Background contamination (Table S3) was monitored by processing operational blanks (unexposed filters) simultaneously with field samples."

**19. Section 2.5**

Consider renaming it as "Data Analysis and Statistical Method.

**Response:**

Suggestion taken.

20. Line 216: The data must be tabulated instead of shown in a figure. The values are more appropriate to demonstrate the actual circumstances.

**Response:**

Thank you for your valuable suggestion. We have added Table S2 in the Supporting Information (SI) file.

21. The values and percentages must be presented in the form of mean plus standard deviation. Check out all these in all sections.

**Response:**

Thank you for your valuable comments. We have revised all values to the format of mean  $\pm$  standard deviation throughout the text.

22. The uses of PM10 and PMcoarse are confusing Standardize to use one term.

**Response:**

Sorry for the confusion. We have standardized our terminology by replacing " $PM_{coarse}$ " with " $PM_{2.5-10}$ " (particulate matter with aerodynamic diameters between 2.5 and 10  $\mu$ m). When characterizing microplastics and plasticizers, we specifically use "fine particles ( $PM_{2.5}$ )" and "coarse particles ( $PM_{2.5-10}$ )" to clearly differentiate and compare their characteristics across different size fractions.

However, we need to retain " $PM_{10}$ " in the sections of the source profiles and health risk assessments of MPs and plasticizers. For the source profile and health risk assessment sections, we primarily use " $PM_{10}$ " because it represents the total inhalable particulate matter that is most

relevant for exposure assessment and it allows direct comparison with other studies.

**23. Line 230: The verb "crushed" is inappropriate.**

**Response:**

This point has been revised as follows.

"One possible explanation for this is that plastic waste can be fragmented into MPs during the process of combustion."

**24. Line 274: What is the meaning of "greenhouses" here?**

**Response:**

Thank you for your question. In original Line 274, the term "greenhouses" refers to controlled agricultural environments, plastic film structures covering crops, which are used to regulate climatic conditions. In this study, AF (agricultural film) primarily refers to greenhouse coverings, which are generally made of transparent polyethylene and are typically less than 8 µm thick (Wang et al., 2018; Li et al., 2022). The compositions of MPs in AF may be influenced by wear/degradation of plastic film and other plastic equipment in greenhouses. The original lines 272-274 suggest that the high PS content in AF could stem from PS-based materials (e.g., lamp-chimneys, electrical devices) commonly used in greenhouse facilities.

To avoid ambiguity, we revised the original sentence (Lines 272-274) as follows:

"Agricultural facilities made of PS (e.g., lamp-chimneys, electrical devices) in greenhouses may influence the MPs composition of Agricultural Film source (Qi et al., 2023)."


**Response:**

Thank you for your comment. We have revised Figure 5 to incorporate legends within the figure. The revised Figure 5 is as follows.

**Figure 5** Source-Pathway-Receptor model associated with three different MP and plasticizer sources.

29. The DTT included in this study was reported in volume-base. What is the reason for choosing this DTT to describe the toxicity of PMs?

**Response:**

We adopted the volume-based DTT assay in this study to maintain consistency with the volume-based quantification of MPs and plasticizers in PM2.5 and PM10 (quantified as mass per cubic meter of air). This approach facilitates a clearer interpretation of the relationship between the concentrations of MPs/plasticizers and the oxidative potential of PMs per unit volume of air (m3).

30. The future research directions must be advanced. For example, it is recommended to consider coupling the multiple ecological health assessment methods mentioned in this study, measure the weights of different methods, and provide comprehensive evaluation indices.

**Response:**

Thank you for your valuable suggestion. We have revised the relevant sentence in the manuscript to incorporate the integration of multiple ecological health assessment methods.

"Future studies should expand the range of assessed MPs and plasticizers and integrating multiple ecological health assessment methods to further refine the health risk assessment system and deepen the understanding of the environmental and health hazards of MPs."

**Reviewer 3:**

Liu et al. investigated the characteristics, source profiles, and health risks of airborne microplastics and plasticizers emitted from plastic burning, fruit bag burning, road traffic, agricultural film, and livestock breeding sources in Guanzhong Plain, Northern China. Knowledge on the characteristics of atmospheric microplastics in Northern China is currently limited. This study provides important insights for source tracking and risk assessment of these emerging contaminants. I recommend this paper to be published after minor revisions.

Dear reviewer,

Thanks very much for taking your time to review this manuscript. We sincerely appreciate your constructive suggestions for improvement. We have carefully considered each of your comments and have revised the manuscript accordingly to improve the quality of our manuscript.

1. Many plasticizers were used in the manufacturing of plastics. The focus of this study is PAEs, BPs and BPA. Please provide the reason for selecting these three groups of plastics to study. Response:

The selection of BTs, PAEs, and BPA as the focus of our study is driven by their widespread use, ubiquitous in the environment and potential health risks.

Phthalate esters (PAEs) are the most widely used plasticizers globally, dominating the plastic additive market. He et al. (2020) demonstrated that during 2007-2017, the annual global production of PAEs increased from 2.7 million tons to 6 million tons. Moreover, China is recognized as the largest importer of PAEs worldwide. Benzothiazoles (BTs) are extensively used in automotive tires and agrochemicals. High concentrations of BTs were discovered in the street runoff, suggesting that these tire material-related compounds can persevere in the environment (Zhang et al., 2018). Exposure to BTs may result in central nervous system depression, liver and kidney damage, dermatitis, and pulmonary irritation (Ginsberg et al., 2011). Bisphenol A (BPA) as a common industrial chemical component in many products, has steadily grown over the last 50 years (Corrales et al., 2015). Growth of global production has consistently ranged between 0% and 5% annually (Corrales et al., 2015). PAEs and BPA considered as endocrine disruptors, are demonstrated to impair reproductive function and development in laboratory animals (Wang et al., 2018).

Despite the well-documented health risks associated with these plasticizers in laboratory settings, there is a significant gap in understanding their real-world emissions and health impacts. In this study, we aim to fill this gap by investigating the emission characteristics of these plasticizers from various sources and evaluating their potential health impacts based on real-world concentration levels.

The related sentences have been revised as follows.

"Plasticizers are widely used in the production of plastics in order to achieve the desired material properties (Demir and Ulutan, 2013). Since plasticizers are not chemically bound to the plastic products, they can easily diffuse into the surrounding environment during the life-time (Yadav et al., 2017; Demir and Ulutan, 2013). PAEs,BTs, and BPA are the most common plastic additives that are ubiquitous in the environment and pose potential health risks. Phthalate esters (PAEs) are the most widely used plasticizers globally, dominating the plastic additive market. He et al. (2020) demonstrated that during 2007-2017, the annual global production of PAEs increased from 2.7 million tons to 6 million tons. Moreover, China is recognized as the largest importer of PAEs

worldwide. Benzothiazoles (BTs) are extensively used in automotive tires and agrochemicals. High concentrations of BTs were discovered in the street runoff, suggesting that these tire material-related compounds can persevere in the environment (Zhang et al., 2018). Exposure to BTs may result in central nervous system depression, liver and kidney damage, dermatitis, and pulmonary irritation (Ginsberg et al., 2011). Bisphenol A (BPA) as a common industrial chemical component in many products, has steadily grown over the last 50 years (Corrales et al., 2015). Growth of global production has consistently ranged between 0% and 5% annually (Corrales et al., 2015). PAEs and BPA considered as endocrine disruptors, are demonstrated to impair reproductive function and development in laboratory animals (Wang et al., 2019).

Previous studies have investigated the emission characteristics of plasticizers from various sources. Simoneit et al. (2005) illustrated that the major plasticizers detected in particulate matters (PMs) from open-burning of plastics were dibutyl phthalate (DBP), diethylhexyl adipate (DEHA), and diethylhexyl phthalate (DEHP). Zeng et al. (2020) reported phthalate concentrations in greenhouses air were higher than that in ambient air. Liu et al. (2023) found that phthalates were the most dominant plasticizer compositions in tunnel PM2.5, accounting for 64.8% of the detected plasticizers. Zhang et al. (2018) demonstrated that tire material-related compounds, benzothiazole (BT) and 2-hydroxybenzothiazole (2-OH-BT) were the major compounds in both tire and road dust samples. The majority of existing studies on atmospheric MPs and plasticizers have focused on analyzing the emission characteristics of individual source and lacked a comprehensive and comparative analysis of the MPs emission profiles of various sources."


  Response:

We did consider the effect of particulate matter particle size in our risk assessment. Specifically, we compared the difference in risk indices for PM10 and PM2.5. As shown in Figure S2, for the same source, the difference in non-carcinogenic risk for different particle sizes is small. This suggests that the effect of particle size on MPs induced health risk is relatively limited. Therefore, in the Discussion section, we focused more on comparing the differences in health risk between sources rather than particle sizes.

**Figure S2** The non-cancer risks of MPs and plasticizers from five sources (PB: Plastic Burning, FB: Fruit-bag Burning, RT: Road Traffic, AF: Agricultural Film, LB: Livestock Breeding).

**5. Line 51, Agricultural activities are also a significant contributor to atmospheric MPs? Response:**

Thank you very much for your valuable comment. Although agricultural activities are not a direct source of microplastics (MPs), they serve as a significant indirect contributor to atmospheric MPs. due to the use of plastic mulch films and other related plastic products, as well as the deposition of MPs in agricultural soils transported via atmospheric and surface runoff pathways, MPs are present in the soil (Lakhiar et al., 2024). Agricultural activities, by increasing soil disturbance, may cause the resuspension of MPs into the atmosphere. In this way, agricultural activities indirectly contribute to atmospheric MPs. For example, the paper "Activities of Microplastics (MPs) in Agricultural Soil: A Review of MPs Pollution from the Perspective of Agricultural Ecosystems" clearly indicates that agricultural machinery operations, such as plowing and harvesting, can disturb the soil and expose MPs to the air. These particles can then be released into the air by wind.


**Response:**

The authors have added the references in Lines 60 and 66.

**Reviewer 5:**

This paper describes the microplastics in the coarse and fine aerosol fractions from five different sources in northern China, quantified using pyrolysis GC/MS. A selection of common plastic additives was also detected. The oxidative potentials of filter extracts were determined using a DTT assay. Livestock breeding and plastic burning were found to produce the largest potential hazards of the five sources tested. This work represents an important study in a growing area of interest in atmospheric science, and I would recommend publication if the authors can add clarity to their methodology, QA/QC protocols, and data analysis.

**Dear reviewer.**

Thanks very much for taking your time to review this manuscript. We have carefully reviewed your comments and made the appropriate revisions to improve the quality of our manuscript. We have already added more details to clarify our methodology, QA/QC protocols, and data analysis. Below are our detailed responses to your comments:

**Specific comments:**

Line 39 "Global plastic production has gradually increased" – has it gradually or exponentially increased?

**Response:**

Thank you for highlighting this important distinction. Based on data from "Production, use, and fate of all plastics ever made (Sci. Adv., 2017, 3, e1700782)", we observed that global plastic production growth accelerated significantly after 1990, with a nearly vertical rise post-2000. Thus, "exponentially increased" more accurately reflects the trend of global plastic production than "gradually increased". We have revised the sentence as follows.

"Global plastic production has increased exponentially after 1990, resulting in serious environmental contamination."


Line 99 – Section 2.1, can the authors provide sampling dates and a map of where sampling took place (with long and lat)? This would be very helpful for future modelling studies.

**Response:**

We have revised the Figure S1 to indicate the sampling locations, sampling dates, and respective longitudes and latitudes for different sources (PB: Plastic Burning, FB: Fruit-bag Burning, RT: Road Traffic, AF: Agricultural Film, LB: Livestock Breeding).

**Figure S1** Sampling sites for different sources (PB: Plastic Burning, FB: Fruit-bag Burning, RT: Road Traffic, AF: Agricultural Film, LB: Livestock Breeding).

Line 121 – what was the material of the air cassettes used? And did it influence the final results?

The air cassettes used in our study were made of glass, a material selected for its chemical inertness to ensure no interference with experimental results. Each cassette was thoroughly rinsed with ethanol absolute before use.

Line 122 – did filter weighing follow any published standard and was static electricity neutralised prior to weighing?

**Response:**

The filter weighing process in our study was conducted in accordance with the Chinese National Standard GB/T 39193-2020 and the reference "Chemical characterization of PM2.5 in heavy polluted industrial zones in the Guanzhong Plain, northwest China: Determination of fingerprint source profiles (Sci. Total Environ., 2022, 840, 156729).", which provide detailed protocols for the determination of particulate matter mass concentration in ambient air, ensuring the reliability of our filter weighing procedures. We used anti-static instrument (ANTIST-kit-un, MettlerToledo, Switzerland) to neutralize static electricity prior to weighing the filters.


"The field blank of each type of source was synchronously collected with active sampling. Unused filters (the same batch as sampling filters) were loaded into identical sampling devices, which were placed adjacent to operational samplers for the entire duration of one sampling event."

Line 125 – Several studies have found that nitrile gloves can potentially influence pyrolysis GC/MS analysis of polymers, particularly PE. Please discuss.

**Response:**

We sincerely appreciate the reviewer's insightful comment. We fully acknowledge this concern and have taken multiple precautions in our experiment to minimize any possible interference.

Firstly, we cleaned the gloves with ethanol absolute before the experiment. This effectively removed potential contaminants from the gloves, minimizing their interference with the pyrolysis GC/MS analysis.

Secondly, we used stainless steel tweezers to handle the filter membrane throughout the experiment, avoiding direct contact between the gloves and the filters. This further ensured the reliability of the analysis results.

Moreover, the blank experiment was also conducted with gloves on, and the results from the blank experiment have been subtracted. This step effectively eliminated the potential interference that

**might be caused by the gloves, ensuring the accuracy of the final results.**

Line 138 – Please provide explicit details of preparation of samples, instrument configurations, and QA/QC protocols in this manuscript rather than referring to another paper. In regards to QA/QC, please provide information on positive and negative controls, the amount of analytes detected in background samples, and any subtractions that were completed to produce the final values. Addition to the supplemental would be a satisfactory placement for this information.

**Response:**

We do agree with the reviewer's comment. We have now incorporated the QA/QC protocols into Section 2.3 of the Methods. Additionally, the detailed sample preparation procedures and instrument configurations are provided in the Supplemental Information. The specific information is as follows:

"Quality assurance/Quality control (QA/QC)

The flow rates of all samplers were calibrated using a mass flowmeter (Model 4140, TSI, Shoreview, MN, USA) before and after each sampling cycle. All quartz filters used in this study were preheated at 800°C for 3 h to remove any potential contaminants and then cooled before use. To minimize experimental error, sampling was conducted in duplicate for each particle size of each source. For the chemical measurements, one in every 10 samples was reanalyzed for quantity assurance purposes, and the SD errors of replicate trials were within 10% for the pyrolysis analyses. Calibration curves were established using reference standards. The linearities of the standard calibration curves were > 0.987. The standard deviations of the pyrolyzed standard were within 94.1% to 98.3%. Background contamination (Table S3) was monitored by processing operational blanks (unexposed filters) simultaneously with field samples."

**"Appendix 1 Analysis of Microplastics**

In the quantification of miccroplastic (MPs) contents, 0.526 cm2 of a filter sample was folded with ferromagnetic pyrofoil (F670, Japan Analytical Industry Co., Ltd, Tokyo, Japan) using a clean flip head tweezer. The pyrofoil was loaded onto a Curie-point pyrolyzer (JHS-3, Japan Analytical Industry Co., Ltd) coupled with a GC/MS (7890GC/5975MS. Agilent Technology) system, and was rapidly heated to 670°C at 5 s. The interface between Curie point and GC injection port was  $300^{\circ}$ C. The pyrolyzed compounds were separated with a DB-5ms capillary column (30 m  $\times$  0.25 mm × 1 µm film thickness; J&W Scientific, USA). The initial the GC oven temperature was at 50°C for 5 min, increase at a rate of 25°C min-1 to 300 °C, and then hold at the final temperature of 310°C for 10 min. The carrier gas was ultra-high purity helium (He). The mass selective detector (5975, Agilent Technology) was at a full scan mode from 30 to 500 amu under electron ionization (EI) at a voltage of 70 eV and an ion source temperature of 230°C. Peaks were identified from the known fragmentation, mass spectra, and retention times of the target pyrolysis products of detected MPs (Table S1). Calibration curves were prepared using the reference standards. All standards excepting rubbers (99%, JSR Corporation) were purchased from Dupont  $(\geq 98\%, USA)$ . The linearities of the standard calibration curves were > 0.987. As markers for butadiene rubber (BR) (e.g., vinylcyclohexene) have interferences from SBR, BR is not quantified independently; instead, the sum of SBR and BR is reported to reduce uncertainties."

Line 140 – please provide manufacturer information on the thermal desorption unit used. Response:

The statement has been revised as follows.

"An in-port thermal tube (78 mm long, 4 mm I.D., 6.35 mm O.D., Agilent Technology, USA) coupled with a GC/MS system (7890GC/5975MS, Agilent Technology, USA) was utilized to analyze phthalates."

Line 142 (and elsewhere) – please provide supplier information and purity for all chemicals used in the study.

**Response:**

The supplier details and purity of all chemicals used in this study have been added to the Methods section.

"An in-port thermal tube (78 mm long, 4 mm I.D., 6.35 mm O.D., Agilent Technology, USA) coupled with a GC/MS system (7890GC/5975MS, Agilent Technology, USA) was utilized to analyze phthalates, including dimethylphthalate (DMP), diethyl phthalate (DEP), di-n-butyl phthalate (DBP), butyl benzyl phthalate (BBP), bis(2-ethylhexyl)phthalate (DEHP), and di-n-octyl phthalate (DnOP) All PAEs were purchased from Sigma-Aldrich ( $\geq$  98%, Steinheim, Germany)." "Nine types of benzothiazole (97%, Thermofisher Scientific Co., LTD, Waltham, MA, United States) related compounds were quantified"

**Line 154 – the extraction and concentration procedures need to be provided in detail. Response:**

Thank you for your comments. The extraction and concentration procedures were performed as detailed in Appendix 2 in Supplemental Information, including ultrasonic extraction and purification.

"The filter is transferred into a test tube, and 10 mL of a mixture of ultrapure deionized water (18 M-Ohm) and methanol (HPLC grade, Fisher Chemical, USA) (5:3, v:v) is added. The sample is extracted in an ultrasonic water bath at room temperature for 60 min. The combined extracts are concentrated and then diluted with ultrapure deionized water containing 0.2% v/v formic acid (~pH 2.5). The diluted extract is purified using an Oasis HLB Flangeless Vac Cartridge (3cc, 60 mg sorbent per cartridge, 30 µm particle size; Waters, USA)."

**Line 158 – please provide method details instead of referring to a different paper. Response:**

We sincerely appreciate your suggestion. As requested, we have now provided the method details in Supplementary Information (Appendix 2). The specific information is as follows:

**"Appendix 2 Analysis of plasticizers**

Phthalates were quantified using in-injection port-thermal desorption/mass spectrometry (TD-GC/MS) method. Aliquots of the filters (1.578 cm²) were cut into small pieces, spiked with ISs (Chrysene-d12, 96%, LGC Standard Limited, United States), and inserted into thermal tubes (78 mm long, 4 mm I.D., 6.35 mm O.D., Agilent Technology, USA) for analyses. The sample tube was directly loaded into a GC injection port (GC7890, Agilent Technology), at an initial temperature of 50°C. The temperature of injector was then ramped to 275°C for desorption in a splitless mode, while the GC oven temperature was kept at 30°C. The desorbed analytes were refocused at the column head. After the injector temperature reaches the set point, the oven program starts. The analytes were speared by an DB-5ms capillary column (30 m × 0.25 mm i.d. × 0.25 µm film

thickness; J&W Scientific). The carrier gas was ultra-high purity (99.9999%) He at a constant flow of 1.0 cm3 min-1. The MSD (5975, Agilent Technology) was full scanned from 50 to 550 amu under electron impact ionization (EI) at a voltage of 70 eV and an ion source temperature of 230°C. Identification was achieved by characteristic ion and retention times of the chromatographic peaks with those of authentic standards.

To quantify benzothiazole and its derivatives, each of 1.578 cm2 of the filter sample was cut and spiked with an internal standard (IS) of benzothiazole-d4 (95%, LGC Standard Limited, United States). The filter is transferred into a test tube, and 10 mL of a mixture of ultrapure deionized water (18 M-Ohm) and methanol (HPLC grade, Fisher Chemical, USA) (5:3, v:v) was added. The sample was extracted in an ultrasonic water bath at room temperature for 60 min. The combined extracts were concentrated and then diluted to with ultrapure deionized water containing 0.2% v/v formic acid (~pH 2.5). The diluted extract was purified using an Oasis HLB Flangeless Vac Cartridge (3cc, 60 mg sorbent per cartridge, 30 µm particle size; Waters, USA). The target analytes were eluted with 5 mL of methanol and the eluents were evaporated to 1 mL under a gentle nitrogen stream prior to analysis. The separation of target analytes was performed using an ultra-performance liquid chromatography (UPLC; ACQUITY, Waters), and both identification and quantification were accomplished using a triple quadrupole mass spectrometer (ESI-MS/MS; Xevo TO-S, Waters). An ACOUITY UPLC BEH SHIELD RP 18 column (100 mm ×3 mm × 1.7 mm) was serially connected to a Vanguard column (BEH C18, 5 mm × 2.1 mm × 1.7 mm). The mobile phase comprises 100% methanol (A) and ultrapure deionized water acidified with 0.1% (v/v) formic acid (B) at a flow rate of 450 mL min-1. A gradient elution program was applied for the separation. The tandem MS system was operated in the positive ion multiple reaction monitoring mode. Identification was achieved by characteristic ion and retention times of the chromatographic peaks with those of authentic standards."

Line 167 – please discuss potential for interferences with detection or any specificity testing that was completed.

**Response:**

In order to avoid fluorescence detector saturation owing to matrix components in the quantification of BPA, the excitation and emission wavelengths from 0 to 14 min were fitted at 210 and 220 nm, respectively. Besides, to eliminate potential interferences from other organic compounds in PM samples, we employed both peak purity testing and spectrum matching in the quantification of BPA. For peak purity, we set a scan threshold of 1 mAU and required a peak coverage of 95%. For spectrum matching, we established a similarity threshold of 0.98. These rigorous measures effectively confirmed the presence of BPA in the samples, free from interference from other compounds.

Line 168 – was quantification completed using an internal standard or external standard method? Response:

Quantification of total bisphenol A (BPA) in this study was completed using an external standard method. Calibration curves for BPA were prepared using BPA standard ( $\geq$  99%, Sigma-Aldrich, Steinheim, Germany) solutions in acetonitrile. The linearities of the standard calibration curves were > 0.987. The recoveries of bisphenol A (BPA) are in a range of 96.3%-99.3%.

Section 2.3. – With regard to the FB filters, presumably many combustion products would be produced that no longer resemble the chemical structure of the original plastic. Please discuss the potential of these compounds to influence the oxidative testing.

**Response:**

Thank you for your valuable comment. Our results demonstrated that PMs from combustion sources exhibited higher oxidative potential (OP) compared to non-combustion sources. This suggests that the combustion process may enhance the OP of PM. This phenomenon may be associated with the products generated from the combustion of plastics. However, as we were unable to isolate these combustion products from PM, we could not quantify the impact of polymer combustion products on the OP results.

In our study, we focused on exploring the relationship between the DTT activity of PMs and microplastics and plasticizers using a correlation method. We focused on the oxidative potential of residual microplastics and plasticizers after combustion, rather than their thermal transformation products.

We acknowledge the importance of investigating the specific contributions of combustion-derived microplastics to PM OP. In future research, we propose conducting DTT assays on standard microplastic subjected to combustion treatment, comparing with non-combusted microplastics. This approach could provide valuable insights into the role of combustion products of microplastics in altering the oxidative potential of PM.

Line 221 – since wax coatings often contain a mixture of alkanes, please discuss any potential interferences of the wax on the pyrolysis GC/MS results (e.g. PE quantification).

**Response:**

The reason we distinguish between plastic burning (PB) and fruit bag burning (FB) rather than classifying them as a single combustion source is that the wax layer in fruit bags cannot be separated from the plastic. This is a featured source in the Guanzhong region (This is also quite common in fruit producing areas in northern China), and local residents typically burn fruit bags directly without separating the wax.

By comparing the results of PB and FB, we found that, there were no significant differences in the composition and concentration of microplastics and plasticizers emitted from PB and FB. These results indicate that the wax coating in FB has minimal influence on the analytical results.

Line 223 – the MPs in PB and LB were 50% of the PM, was this by mass? Please clarify and also present the percentage of PM that was attributed to plastic in all scenarios.

**Response:**

We apologize for causing confusion to the reviewer. The 50% mentioned in original Line 223 refers to the relative proportion of coarse and fine microplastics from PB and LB, meaning that the combined mass percentage of coarse and fine microplastics is 100%. The sentence has been revised as follows.

"Notably, MPs in Plastic Burning and Livestock Breeding constituted a comparable proportion in both fine and coarse fractions, both close to 50%."

We also presented the mass proportion of microplastics in PMs from five sources in Figure S4.

**Figure S4** The mass proportion of MPs in PMs from five sources (PB: Plastic Burning, FB: Fruit-bag Burning, RT: Road Traffic, AF: Agricultural Film, LB: Livestock Breeding).

Line 226 – how was the reported uncertainty calculated? Response:

At each sampling location, we collected 2-3 replicate samples for each particle size fraction. The standard deviation was calculated to represent the uncertainty.

Line 261 – Please discuss the details of the flyover? How was this sample collection done? Response:

The sampling was conducted on a pedestrian flyover (height: ~4 m) near the Xi'an Jiaotong University campus in Beilin District, Xi'an. The flyover spans an urban arterial road with four motorized lanes (bidirectional), two dedicated bicycle lanes. The traffic volume is substantial, with private cars being the primary type of vehicle. We did not choose to conduct sampling on the ground by the roadside in order to avoid direct interference from road dust and vehicle exhaust.

The MiniVOL samplers (Airmetrics, Springfield, OR, USA), inertial impactors, was conducted on the flyover railing (see the following photo). In order to prevent any potential interference from pedestrians on the flyover, we cordoned off an area of 1.5 meters around the sampler with caution tape and affixed signs indicating that the area was designated for scientific research purposes.

Line 394 – please cite where these compounds are accepted as carcinogens.

**Response:**

We have revised the sentence to include the necessary citations to support this statement.

"ILCR for the three carcinogenic compounds (BT, BBP, and DEHP) were calculated in this study (Guyton et al., 2009; Ma et al., 2020; Liu et al., 2023)."


"The aims of this research are to characterize the distributions of MPs and plasticizers in dual-size PMs ( $PM_{2.5}$ ,  $PM_{2.5-10}$ ) from typical MP sources (anthropogenic sources from daily life) in the Guanzhong Plain."

The reason we distinguish between plastic burning (PB) and fruit bag burning (FB) rather than classifying them as a single combustion source is that the wax layer in fruit bags cannot be separated from the plastic. This is a featured source in the Guanzhong region (This is also quite common in fruit producing areas in northern China), and local residents typically burn fruit bags directly without separating the wax. We have revised this point in the manuscript as follows.

"The Guanzhong Plain is an important fruit production base in China, with the highest

consumption of fruit bags. Local residents often use the above-mentioned plastic products to ignite solid fuels for indoor heating or cooking. The reason we distinguish between Plastic Burning and Fruit-bag Burning rather than classifying them as a single combustion source is that the wax layer in fruit bags cannot be separated from the plastic. This is a featured source in the Guanzhong region (This is also quite common in fruit producing areas in northern China), and local residents typically burn fruit bags directly without separating the wax. Table 1 provides a summary of the essential details for each source."

The more details of fruit bags and their differences with household plastic waste have been added in the manuscript as follows.

"Fruit bags are typically lightweight, thin-film, which are designed for single-use and are often discarded after a short period, differing from common household plastic waste. These bags are usually made from low-density polyethylene and Nylon, which is known for its flexibility and transparency (Ali et al., 2021; Yang et al., 2022)."


**Response:**

Thank you for your suggestion. We have revised the manuscript to remove abbreviations of PB, FB, RT, AF, LB, and OP in the main text to enhance readability. However, in order to maintain conciseness and adhere to the word limit, we have retained the abbreviations in the Abstract and Figures/Tables.

**Specific comments:**

■ Line 89: Can you elaborate why is the UV so strong in this area and why does it matter for this study? MPs emitted in this area may undergo increased photooxidation, is that something that can be discussed with your results? Or is that something that needs to be explored further in future studies?

**Response:**

The Guanzhong Plain, our study area is located in northwestern China, and according to the study "Ultraviolet radiation over China: Spatial distribution and trends (Sust. Energ. Rev., 76, 1371-1383)", this region experiences relatively strong UV radiation (14486 kJ m-2 day-1, the average solar radiation of Xi'an, Tongchuan, and Xianyang). The high UV radiation in this area can be attributed to the elevated altitude (497 m, the average elevated altitude of Xi'an, Tongchuan, and Xianyang) (reducing atmospheric path length) and less cloud cover (especially low cloud cover). Ultraviolet radiation drives photooxidation of plastics, breaking polymer chains and accelerating fragmentation into secondary MPs. The high UV intensity in the Guanzhong Plain exacerbates this process.

However, our current research primarily focused on the characteristics and health risks of MPs and plasticizers emitted from various sources in the region. We did not specifically examine how UV-induced aging of MPs might influence their environmental behavior or toxicity. This aspect is important as ultraviolet radiation can alter the physical and chemical properties of MPs, potentially increasing their toxicity and capacity to adsorb other pollutants. We accept the reviewer's suggestion and plan to explore the effects of UV on MPs and their associated health

risks in future studies. This will provide a more comprehensive understanding of the eco-health impacts of MPs in the Guanzhong Plain.

**Reference:**

Liu, H., Hu, B., Zhang, L., Zhao, X. J., Shang, K. Z., Wang, Y. S., and Wang, J.: Ultraviolet radiation over China: Spatial distribution and trends, Renew. Sust. Energ. Rev., 76, 1371-1383, 10.1016/j.rser.2017.03.102, 2017.

■ Line 113: State what type of sampler MiniVol samplers are (e.g. impinger, impactor, etc.) Response:

Thank you for this comment. The MiniVol air samplers used in this study are impactors. This point has been revised as follows.

"MiniVOL samplers (Airmetrics, Springfield, OR, USA), which are inertial impactors, were employed for collection, operating at a steady flow rate of 5 L min-1."

■ Figure 1: errors bars used represent standard deviation of PM10 however the figure is plotting PM5 and PMcourse. Is that error representation suitable for this plot? Consider just focusing on PM2.5 and PMcourse throughout the paper and removing PM10 to be more consistent with how results are discussed. Is that error representation suitable for this plot? Consider just focusing on PM2.5 and PMcourse throughout the paper and removing PM10 to be more consistent with how results are discussed.

**Response:**

We have revised Figure 1 to illustrate the characteristics of microplastics and plasticizers in  $PM_{2.5}$  and  $PM_{2.5-10}$  ( $PM_{coarse}$ ) and remove  $PM_{10}$  from this figure. Additionally, we have included error bars representing the standard deviation for both  $PM_{2.5}$  and  $PM_{2.5-10}$ .

**Figure 1** Average concentrations of MPs (a) and plasticizers (b) in PM2.5 and PM2.5-10 from five sources (PB: Plastic Burning, FB: Fruit-bag Burning, RT: Road Traffic, AF: Agricultural Film, LB: Livestock Breeding).

Figure 1: I would suggest increasing the spacing between (a) and (b) to improve readability. The secondary y-axis for (a) is very close the first y-axis of (b).

**Response:**

Suggestion taken. The figure was shown as above.

■ Line 282: In addition to the most abundant plasticizer, I would suggest the authors list the next two most abundant plasticizer types detected in PB.

We apologize for causing confusion to the reviewer. It has been revised as follows. "The highest concentrations of PAEs, BTs, and BPA still appeared in PB among five sources."

■ Line 288-289 and Figure 5: Use consistent spelling of tire or tyre throughout the paper.

**Response:**

Sorry for the error. We have revised line 288-289 and Figure 5 to consistently use the spelling "tire" throughout the revised manuscript.

■ Figure 3: Redefine abbreviations when they appear in captions e.g., PAE and BT Response:

Suggestion taken.

■ Figure 4: Consider adding visual cues (e.g. labels by the arrow, highlighted section or color cues) to help guide the reader's eye from the black arrow to the corresponding source marker on the x-axis. In the main text, please expand the discussion for these identified markers. How confident are these marker classifications? Have previous studies reported the same markers?

**Response:**

We have added the abbreviation of the fingerprint species next to the arrow in Figure 4 to highlight the substance. The revised Figure is as follows.

**Figure 4** Source profiles of microplastics and plasticizers in PM2.5 and PM10 (The black arrows indicate the source markers).

The marker classifications in this study were based on the distinct emission characteristics of different sources. We compared and selected these markers to represent each source accurately based on our real-world experiment results. While previous studies on the emission characteristics of microplastics from these sources are relatively limited, there are some findings that corroborate our results. We have expanded the discussion for the identified markers in the manuscript as follows.

"Liu et al. (2019) documented that polystyrene (PS) were the predominant polymer in agricultural sources."

"Liu et al. (2023) revealed that natural rubber (NR) and other rubber particles were emitted at high levels in tunnel traffic, emerging as the dominant microplastic in traffic-dominated environments."

■ Figure 6: Please add a legend to define the colour groupings.

**Response:**

Suggestion taken.

Figure 6 Correlations between DTT, MPs, and plasticizers (\*P < 0.05; \*\*P < 0.01).

**Technical corrections:**

■ Line 19: (phthalates, benzothiazole and its derivatives, and bisphenol A)

**Response:**

Suggestion taken. It has been revised as follows.

"the characteristics and source profiles of eight types of common MPs and three classes of plasticizers (i.e., phthalates, benzothiazole and its derivatives, and bisphenol A)."

■ Line 31: missing closing bracket: poly(methyl methacrylate)

**Response:**

Corrected.

■ Line 122: missing space: ...frozen at -20°C

Response:

Corrected.

■ Line 180: and absorbance was measured at 412 nm using a microplate reader

Response:

Corrected.

■ Line 246: This is because plastic products...

Response: Corrected.

---

## Author Response (AR2)

Upon review, the authors made changes that have greatly improved the manuscript. My only comment relates to the impacts of choosing not to use a sample pretreatment step prior to Py GCMS analysis.

The identification of airborne polymers using direct injection thermal techniques has been used previously, although this typically involves a double shot method that employs a thermal desorption step prior to pyrolysis (e.g. Mizuguchi et al., 2023; Chen et al., 2024; Gregoris et al., 2023). For complex samples containing suspected combustion products, it is uncertain if quantification can be performed without the use of a pretreatment step, as to my knowledge, no method-focused studies to date have been reported for these types of aerosol samples. That being said, the ability to identify polymers will rely on the specificity of the pyrolysis markers used in the Py GCMS analysis (Gregoris et al., 2023). The markers used for quantification in this study are not commonly reported in combustion samples, with the possible exception of those used for PVC quantification.

The authors report using quantification markers 1-chloroindan and dihydronaphthaleneazelene for PVC. It is not clear what the structure of these compounds are and whether or not they are applicable to PVC quantification. Chlorobenzene is a known pyrolysis product of PVC (e.g. Yang et al., 2023) and benzyl chloride has also been reported as a possible secondary reaction product between PET and PVC (Coralli et al., 2022), yet I am not familiar with the 1-chloroindan and dihydronaphthaleneazelene reported in this study. The authors should provide CAS numbers for these chemicals and also reference relevant studies that have used these markers. Since compounds similar to chlorobenzene can be produced from combustion processes, there could be a possibility of interferences in the resulting analysis without the use of a thermal pretreatment step.

I think that further discussion on the limitations of the method with regards to the lack of a pretreatment step would be warranted in this manuscript.

**Response:**

**Dear reviewer,**

We sincerely appreciate the time and effort you have dedicated to reviewing our manuscript. We have carefully addressed your suggestion and implemented the necessary revisions. Below are our detailed responses to your comments.

In this study, a single-shot pyrolysis protocol was adopted to markedly reduce analytical time while ensuring quasi-instantaneous and homogeneous thermal pyrolysis (Seeley and Lynch, 2023). This approach has been extensively applied to quantify microplastics in complex environmental samples (Watteau et al., 2018; Becker et al., 2020; Matsui et al., 2020; Lou et al., 2022; Santos et al., 2023). As the reviewer pointed out, when investigating combustion-derived particles, an additional thermal-desorption step would liberate volatile species adsorbed on the particle surface, thereby allowing a finer separation of emitted contaminants and improving accuracy. Consequently, we have explicitly addressed both the strengths and limitations of the single-shot pyrolysis strategy in the Conclusion section, and we will undertake a systematic comparison between the two approaches to Pyrolysis - GC/MS in future work. The related sentences have

been revised as follows.

"The single-shot pyrolysis protocol used in detecting MPs in this study enables quasi-instantaneous and homogenous pyrolysis (Seeley and Lynch, 2023). However, the complex products of combustion process may lead to interference from non-polymeric components during single pyrolysis. Therefore, future work should systematically compare different pyrolysis approaches in order to improve accuracy of MPs for complex samples."

Regarding the qualification markers of PVC, we apologize for causing confusion to the reviewer. The markers referred to as "1-chloroindan" and "dihydronaphthaleneazelene" correspond to three compounds: 1-chloroindan (C9H9Cl, CAS 35275-62-8), 1,2-dihydronaphthalene (C10H10, CAS 447-53-0) and azulene (bicyclo[5.3.0]decapentaene) (C10H8, CAS 275-51-4). Previous studies often used naphthalene, fluorene and other polycyclic aromatic hydrocarbons (PAHs) compounds as PVC markers (Santos et al., 2023; Seeley and Lynch, 2023). However, according to research by Simoneit et al. (2005), PAHs are one of the major products of plastic combustion. Therefore, based on the characteristic pyrolysis products of standard (PVC material), we selected these specific compounds to achieve accurate quantification of PVC in our research.

**Reference:**

- Becker, R., Altmann, K., Sommerfeld, T., and Braun, U.: Quantification of microplastics in a freshwater suspended organic matter using different thermoanalytical methods? outcome of an interlaboratory comparison, J. Anal. Appl. Pyrolysis, 148, 6, 10.1016/j.jaap.2020.104829, 2020.
- Lou, F. F., Wang, J., Sun, C., Song, J. X., Wang, W. L., Pan, Y. H., Huang, Q. X., and Yan, J. H.: Influence of interaction on accuracy of quantification of mixed microplastics using Py-GC/MS, J. Environ. Chem. Eng., 10, 10, 10.1016/j.jece.2022.108012, 2022.
- Matsui, K., Ishimura, T., Mattonai, M., Iwai, I., Watanabe, A., Teramae, N., Ohtani, H., and Watanabe, C.: Identification algorithm for polymer mixtures based on Py-GC/MS and its application for microplastic analysis in environmental samples, J. Anal. Appl. Pyrolysis, 149, 9, 10.1016/j.jaap.2020.104834, 2020.
- Santos, L., Insa, S., Arxé, M., Buttiglieri, G., Rodríguez-Mozaz, S., and Barceló, D.: Analysis of microplastics in the environment: Identification and quantification of trace levels of common types of plastic polymers using pyrolysis-GC/MS, MethodsX, 10, 9, 10.1016/j.mex.2023.102143, 2023.
- Seeley, M. E. and Lynch, J. M.: Previous successes and untapped potential of pyrolysis-GC/MS for the analysis of plastic pollution, Anal. Bioanal. Chem., 415, 2873-2890, 10.1007/s00216-023-04671-1, 2023.
- Simoneit, B. R. T., Medeiros, P. M., and Didyk, B. M.: Combustion products of plastics as indicators for refuse burning in the atmosphere, Environ. Sci. Technol., 39, 6961-6970, 10.1021/es050767x, 2005.
- Watteau, F., Dignac, M. F., Bouchard, A., Revallier, A., and Houot, S.: Microplastic Detection in Soil Amended With Municipal Solid Waste Composts as Revealed by Transmission Electronic Microscopy and Pyrolysis/GC/MS, Front. Sustain. Food Syst., 2, 14, 10.3389/fsufs.2018.00081, 2018.